# Path-conditioned training: a principled way to rescale ReLU neural networks

Arthur Lebeurrier [1]   Titouan Vayer [2]   Remi Gribonval [3]

## Abstract

Despite recent algorithmic advances, we still lack principled ways to leverage the well-documented rescaling symmetries in ReLU neural network parameters. While two properly rescaled weights implement the same function, the training dynamics can be dramatically different. To offer a fresh perspective on exploiting this phenomenon, we build on the recent path-lifting framework, which provides a compact factorization of ReLU networks. We introduce a geometrically motivated criterion to rescale neural network parameters whose minimization leads to a conditioning strategy that aligns a kernel in the path-lifting space with a chosen reference. We derive an efficient algorithm to perform this alignment. In the context of random network initialization, we analyze how the architecture and the initialization scale jointly impact the output of the proposed method. Numerical experiments illustrate its potential to speed up training.

## 1. Introduction

Deep learning has become the dominant paradigm in machine learning, with neural networks achieving breakthrough performance across vision, language, and scientific domains. Training these networks represents a central computational challenge, particularly for large language models where the cost approximately doubles every 6 months (Sastry et al., 2024). Addressing this challenge requires progress

on two fronts: (1) a better understanding of the optimization dynamics, and (2) translating these insights into practical improvements for state-of-the-art models.

Among all neural network architectures, ReLU networks play a singular role, as they have demonstrated strong empirical performance over the years on a lot of tasks (Krizhevsky et al., 2012; Szegedy et al., 2015; Howard et al., 2017; Silver et al., 2016; Dosovitskiy et al., 2021). Importantly, ReLU networks exhibit a well-documented symmetry: specific weights rescaling leave the network function unchanged due to the positive homogeneity of the ReLU activation. This symmetry has many implications.

First, it has proven useful to understand existing optimization algorithms through conservation laws and implicit biases (Du et al., 2018; Kunin et al., 2021; Marcotte et al., 2023; Zhao et al., 2023). In short, these analyses reveal conserved quantities, arising for example from the rescaling symmetry, that constrain and shape the geometry of learning trajectories in both parameter and function space. These effects are closely related to the so-called "rich feature regime" in which the network learns "meaningful" representations from the data (Kunin et al., 2024; Dominé et al., 2025). Moreover, it has been shown that the implicit bias of gradient-based optimization can be characterized through a mirror-flow reparameterization of the dynamics (Gunasekar et al., 2018; Azulay et al., 2021), which can be understood, in part, by leveraging the rescaling symmetry of ReLU networks (Marcotte et al., 2025).

Second, rescaling symmetry has enabled the design of novel optimization methods that exploit this invariance to either accelerate training or reach better local minima. Notable examples include $\mathcal{G}$-SGD (Meng et al., 2019), Path-SGD (Neyshabur et al., 2015), and Equinormalization (Stock et al., 2019), among others (Badrinarayanan et al., 2015; Saul, 2023; Mustafa & Burkholz, 2024). Broadly, these methods can be categorized into two groups: *teleportation* approaches (Zhao et al., 2022; Armenta et al., 2023), which perform a standard optimization step followed by a "symmetry-aware" correction that preserves the implemented function, and algorithms that are intrinsically invariant to the underlying symmetries.

However, the problem of leveraging rescaling symmetries to better understand and improve training dynamics remains

---

[1]ENS de Lyon, CNRS, Inria, Université Claude Bernard Lyon 1, LIP, UMR 5668, 69342, Lyon cedex 07, France [2]Inria, Rennes, France [3]Inria, CNRS, ENS de Lyon, Université Claude Bernard Lyon 1, LIP, UMR 5668, 69342, Lyon cedex 07, France. Correspondence to: Arthur Lebeurrier <arthur.lebeurrier@ens-lyon.fr>.

*Proceedings of the 43$^{rd}$ International Conference on Machine Learning*, Seoul, South Korea. PMLR 306, 2026. Copyright 2026 by the author(s).

open, as the previously mentioned schemes are often guided more by practical considerations than by formal theoretical principles. In this context, the objective of this article is to leverage the concept of path-lifting $\Phi$ (Neyshabur et al., 2015; Bona-Pellissier et al., 2022; Stock & Gribonval, 2023; Gonon et al., 2023) in a geometrically motivated manner to design algorithms that improve training dynamics. The path-lifting $\Phi(\theta)$ provides an intermediate representation between the finite-dimensional parameter space $\theta$ (weights and biases) and the infinite-dimensional space of network realizations $f_\theta$ (the functions implemented by the network).

**Contributions and outline.** After developing a geometric perspective based on path-lifting that clarifies the role of rescaling symmetries in neural network training dynamics, we propose a new geometry-aware rescaling criterion to "teleport" any given parameter $\theta$ to a rescaling-equivalent[1] one $\theta'$ with better training properties, as well as an efficient algorithm to compute $\theta'$, called `PathCond`. We provide a thorough analysis of the effects of our criterion, establishing regimes of interest in terms of initialization scale and architectural choices. We demonstrate that on specific datasets and architectures, using our (parameter-free) rescaling criterion at initialization alone can significantly improve training dynamics, and never degrades it. On CIFAR-10, we show that we can achieve the same accuracy as the baseline in $1.5\times$ fewer epochs.

**Notations.** The data dimension will be denoted by $d$, the output dimension by $k$, parameters of the neural network will be in $\mathbb{R}^p$ and the lifted representation in $\mathbb{R}^q$. The networks will have $H$ neurons.

## 2. Sketch of the idea on a simple example

To illustrate the overall approach considered in this paper, let us start with a standard problem where we have access to $n$ training data $(x_i, y_i)$ with $x_i \in \mathbb{R}^d, y_i \in \mathbb{R}^k$, and, given a loss we wish to find a parametrized function (typically a neural network) $f_\theta : \mathbb{R}^d \to \mathbb{R}^k$ with $\theta \in \mathbb{R}^p$ that minimizes $L(\theta) := \frac{1}{n} \sum_{i=1}^n \text{loss}(f_\theta(x_i), y_i)$. To give intuition about our method, we place ourselves in the idealized scenario where the optimization is carried via a gradient flow

$$\dot{\theta} = -\nabla L(\theta). \tag{1}$$

**From parameter space to lifted space.** Consider the simplest ReLU setting where $H = d = k = 1$ and $f_\theta$ is a one-neuron ReLU network with bias $f_{\theta=(u,v,w)}(x) = u \, \text{ReLU}(vx + w)$. The rescaling invariance property manifests itself via the fact that, for each $\lambda > 0$, the rescaled parameter $\theta^{(\lambda)} := (\frac{1}{\lambda} u, \lambda v, \lambda w)$ satisfies $f_\theta = f_{\theta^{(\lambda)}}$. This function rewrites as $f_\theta(x) = u\mathbf{1}_{vx+w>0}(vx + w)$, that is,

---

[1] so that $f_{\theta'} = f_\theta$

$f_\theta(x) = \mathbf{1}_{vx+w>0}\langle\Phi(\theta), \begin{pmatrix} x \\ 1 \end{pmatrix}\rangle$ where $\Phi(\theta) = (uv, uw)^\top$. One important property of the vector $\Phi(\theta)$, called the path-lifting (formal definition in Section 3), is its rescaling invariance property: $\Phi(\theta) = \Phi(\theta^{(\lambda)})$ for any $\lambda > 0$. In other words, if we rescale the input parameters by $\lambda > 0$ and the output parameter by $1/\lambda$ we do not change $f_\theta$ and $\Phi(\theta)$. In particular for a parameter vector $\theta = (u, v, w)$ with $u > 0$, rescaling with $\lambda := u$ yields that for every input $x$ we have $f_\theta(x) = f_{\theta^{(u)}}(x) = \text{ReLU}(\langle\Phi(\theta), \begin{pmatrix} x \\ 1 \end{pmatrix}\rangle)$, i.e., $f_\theta$ only depends on[2] $\Phi(\theta)$.

With $\theta_{\text{opt}} = (1, 1, 0)^\top$, $f_{\theta_{\text{opt}}} = \text{ReLU}$, and if we sample $x_i \sim \mathcal{N}(0, 1)$, $y_i = f_{\theta_{\text{opt}}}(x_i) = \text{ReLU}(x_i)$, and train with the square loss, the previous discussion leads to two conclusions. First, the risk $L(\theta)$ only depends on $\Phi(\theta)$ when $u > 0$, it can be factorized as $L(\theta) = \ell(\Phi(\theta))$. Second, any model $f_\theta$ satisfying $\Phi(\theta) = \Phi(\theta_{\text{opt}}) = (1, 0)^\top$ implements the same function $\text{ReLU}$ and achieves zero training loss. This motivates an approach in which the learning problem is viewed not only as optimization over $\theta$ in parameter space, *but also as optimization in the lifted space over $\Phi(\theta)$, where targeting $\Phi_{\text{opt}} := \Phi(\theta_{\text{opt}})$ provides a principled strategy from the perspective of minimizing the training loss.*

**Rescaling to mimic gradient descent in lifted space.** The overall approach of this paper is thus guided by a geometric view of the training dynamics *in a lifted representation of the network.* This can be illustrated on our simple example by minimizing $L(\theta)$ with standard gradient descent (GD) and small learning rates (simulating the gradient flow (1)), and observing it both in parameter space and in lifted space, for different initializations (Figure 1). By the chain rule, the ODE (1) also induces an ODE in the lifted space: denoting $P_\theta := \partial\Phi(\theta)\partial\Phi(\theta)^\top$ the *path-kernel*,

$$\partial_t\Phi(\theta) := \frac{d}{dt}\Phi(\theta(t)) = -P_\theta \nabla_\Phi \ell(\Phi(\theta)). \tag{2}$$

The rescaling invariance of $\Phi(\theta)$ *does not extend to the path kernel $P_\theta$*: the latter varies when $\theta$ is replaced by $\theta^{(\lambda)}$. As described later, given $\theta$, `PathCond` aims to identify a parameter rescaling $\lambda$ such that $\theta' := \theta^{(\lambda)}$ somehow *conditions* the path kernel $P_{\theta'}$, i.e. best aligns it with the identity so that $P_{\theta'} \approx I$. This alignment aims to induce trajectories in lifted space close to those that would have been achieved by directly performing gradient descent / flow in the lifted space $\dot{\Phi} \approx -\nabla\ell(\Phi)$, without ever incurring the cost of pseudo-inverting large matrices appearing in "natural gradient" approaches (see discussion in Appendix B).

For our example, such rescaling factors are computed using `PathCond` for three considered initializations, and

---

[2] In general, $f_\theta(x) = \text{sign}(u) \, \text{ReLU}(\text{sign}(u)\langle\Phi(\theta), (x, 1)^\top\rangle)$, so $f_\theta$ only depends on $\text{sign}(\theta), \Phi(\theta)$.

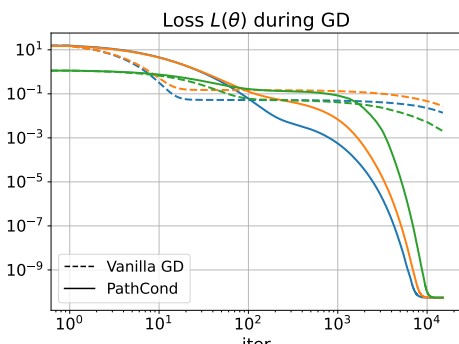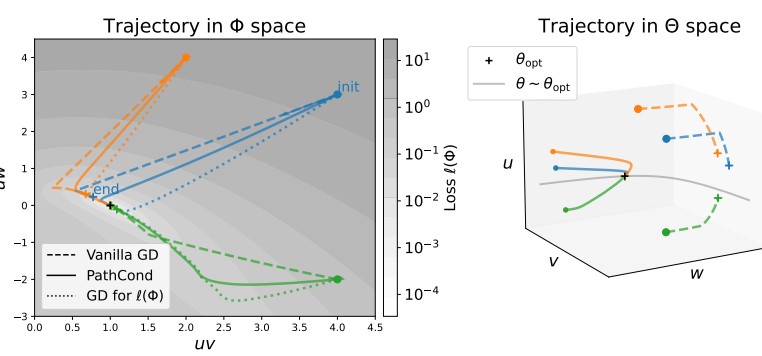

*Figure 1.* GD for a toy model $f_{\theta=(u,v,w)}(x) = u \operatorname{ReLU}(vx+w)$ on a loss $L(\theta)$ that can be factorized as $L(\theta) = \ell(\Phi(\theta))$ (see Section 2). **(Left)** Loss $L(\theta)$ during GD iterations for three different initializations $\theta_0$ (three colors). Dashed lines correspond to GD starting at $\theta_0$, bold lines to GD starting at rescaled $\theta_0^{(\lambda)} \sim \theta_0$ using `PathCond`; **(Middle)** Trajectories in lifted space $\Phi(\theta) = (uv, uw)^\top$. Dotted lines are trajectories corresponding to $\partial_t \Phi = -\nabla_\Phi \ell(\Phi)$ (GD on $\ell(\Phi)$). **(Right)** Trajectories in parameter space.

the resulting trajectories of $(\theta(t), \Phi(\theta(t)))$ are illustrated in Figure 1. As shown in the left subplot, rescaled initializations (plain lines) lead to faster convergence, and indeed, as illustrated in the middle subplot, the trajectories in lifted space obtained with rescaled initializations (bold lines) more closely follow the trajectory of the idealized ODE $\partial_t \Phi = -\nabla_\Phi \ell(\Phi)$ (dotted lines) than the trajectories starting from the non-rescaled initializations (dashed lines).

**Relation to (neural tangent) kernel conditioning.** The general approach described in the next section builds on the vision that the insights drawn from the above simple example, as well as the underlying motivations, extend to larger-scale models. Before entering into the details, let us briefly explain how the resulting approach also bears connections with the conditioning of the celebrated neural tangent kernel (NTK) (Jacot et al., 2018).

As described in many contexts, the dynamic in (1) also induces an ODE on $f_{\theta(t)}(x)$ (at any point $x$) driven by the NTK $K_\theta(x, x') := \partial_\theta f_\theta(x) \partial_\theta f_\theta(x)^\top \in \mathbb{R}^{k \times k}$ (Jacot et al., 2018) where $\partial_\theta f_\theta(x) \in \mathbb{R}^{k \times p}$ denotes the Jacobian of $\theta \to f_\theta(x)$. The NTK can be used to analyze the dynamics of neural networks in specific training regimes. In particular, the spectrum of the NTK governs the convergence rate of the training loss (Bowman, 2023); for instance, larger NTK eigenvalues in certain over-parameterized (lazy) regimes—corresponding to linearized networks (Chizat et al., 2019)—lead to faster convergence rates (Jacot et al., 2018; Arora et al., 2019).

As already observed by Gebhart et al. (2021); Patil & Dovrolis (2021), when applying the general path-lifting framework to ReLU networks, the NTK admits a factorization that separates the architectural contribution from a data-dependent term:

$$K_\theta(x, x') = Z(x, \theta) \, P_\theta \, Z(x', \theta)^\top, \qquad (3)$$

where $P_\theta = \partial\Phi(\theta)\partial\Phi(\theta)^\top$ is the previously introduced

path-kernel and $Z(x, \theta)$ depends on the activations at every ReLU neuron. Equation (3) suggests that modifying the spectral properties of $P_\theta$, for instance through appropriate changes on $\theta$, can improve the training dynamics, and that such modifications can be performed solely based on the network architecture via $\Phi(\theta)$, i.e., *independently of the data*. For example, in the context of parameter pruning, Patil & Dovrolis (2021) exploit this decomposition to select a subset of parameters that maximizes the trace of $P_\theta$ within the subnetwork. In contrast, our approach `PathCond` focuses on improving the conditioning of $P_\theta$ *without altering the function $f_\theta$*, by applying an efficiently chosen rescaling based on a principled criterion.

## 3. Rescaling symmetries and the path-lifting

In this section, we introduce the tools underlying our method `PathCond`, which builds upon the rescaling symmetries of ReLU networks and the path-lifting framework.

**Rescaling equivalent parameters.** As advertised in Section 2, for any neuron in a ReLU network, we can scale its incoming weights by a factor $\lambda > 0$ and simultaneously scale its outgoing weights by $1/\lambda$ without changing the neuron's input-output mapping (Neyshabur et al., 2015; Dinh et al., 2017). This corresponds to multiplying the parameter vector $\theta$ by a certain *admissible* diagonal matrix $D \in \mathcal{D}$ with positive entries. Formally, denoting $\mathcal{D}$ the multiplicative group generated by the set of all such diagonal matrices (when both the chosen neuron and the factor $\lambda$ vary), two parameter vectors $\theta$ and $\theta'$ are said to be rescaling equivalent (denoted $\theta' \sim \theta$) if there is $D \in \mathcal{D}$ such that $\theta' = D\theta$ (see details in Appendix D). This defines an equivalence relation on the parameter space, partitioning it into equivalence classes. Importantly, when $\theta, \theta'$ belong to the same class we have $f_{\theta'} = f_\theta$.

**Gradient flow is not rescaling invariant.** Crucially,

gradient flow does *not* respect this symmetry: rescaling-equivalent parameters $\theta \sim \theta'$ used at initialization of the gradient flow (1) generally lead to different trajectories. This lack of invariance has profound implications. For instance, unbalanced parameter initializations—obtained by applying extreme rescalings (with $\lambda \gg 1$ or $\lambda \ll 1$ for each neuron) to a balanced initialization—can lead to arbitrarily poor optimization and generalization performance (Neyshabur et al., 2015). Conversely, this sensitivity can be exploited: by carefully controlling the rescaling structure, we can hope to guide the training dynamics toward more favorable trajectories (as illustrated in Figure 1). While these ideas have been exploited through several heuristics, which we will review below, our goal is to design a more *principled* heuristic.

**Path-lifting.** The path lifting framework (Neyshabur et al., 2015; Bona-Pellissier et al., 2022; Gonon et al., 2023) factors out the parameter redundancy induced by rescaling symmetries. This is achieved through a lifting map $\Phi$ from $\mathbb{R}^p$ to a lifted space $\mathbb{R}^q$ that captures the essential geometric structure of the network while eliminating redundant degrees of freedom. For any parameter $\theta$ and path[3] p, the p-th entry of $\Phi(\theta)$ writes $\prod_{i \in p} \theta_i$, i.e., is the product of the weights along the path. Interestingly, for any output neuron $v_{\text{out}}$ of $f_\theta$, we can compactly write

$$v_{\text{out}}(\theta; x) = \left\langle \Phi(\theta), A_{v_{\text{out}}}(\theta, x) \begin{pmatrix} x \\ 1 \end{pmatrix} \right\rangle, \qquad (4)$$

where $A_{v_{\text{out}}}(\theta, x)$ is a binary-valued matrix capturing the data-dependent activation patterns (this generalizes the toy example of Section 2). Two of the key properties of $\Phi$ are: a) rescaling-equivalent parameters have the same lifted representation:

$$\Phi(\theta) = \Phi(D\theta), \quad \forall D \in \mathcal{D}. \qquad (5)$$

and b) it induces a (local) factorization of the global loss[4].

$$\forall \theta \in \Omega \subset \mathbb{R}^p, \ L(\theta) = \ell(\Phi(\theta)), \qquad (6)$$

for some $\ell : \mathbb{R}^q \to \mathbb{R}$, enabling the analysis of the dynamics in the lifted space rather than the redundant parameter space (Stock & Gribonval, 2023; Marcotte et al., 2025).

## 4. Path-conditioned training

Using the local factorization of the loss (6), the starting point of our analysis is that, by the chain rule, the variable $z(t) := \Phi(\theta(t))$ in lifted space $\mathbb{R}^q$ satisfies

$$\dot{z}(t) = \partial\Phi(\theta(t))\dot{\theta}(t) = -\partial\Phi(\theta(t))\partial\Phi(\theta(t))^\top \nabla\ell(z(t))$$
$$= -P_{\theta(t)}\nabla\ell(z(t)) \qquad (7)$$

---

[3]A sequence of edges from some neuron to an output neuron.

[4]In the simple example given in Section 2, one possible choice of $\Omega$ corresponds to the region $u > 0$.

as soon as the variable $\theta(t)$ in parameter space $\mathbb{R}^p$ satisfies (1). The *path-kernel* $P_\theta = \partial\Phi(\theta)\partial\Phi(\theta)^\top \in \mathbb{R}^{q \times q}$ (Gebhart et al., 2021; Marcotte et al., 2025) is thus a (local) metric tensor that encodes how parameter updates propagate in the lifted space $\mathbb{R}^q$. This shows that (1), the gradient flow in parameter space $\mathbb{R}^p$, induces a preconditioned flow in the lifted space $\mathbb{R}^q$, with preconditioner equal to $P_\theta$. This can be put in perspective with natural gradient approaches (Amari, 1998; Martens, 2020), where the geometry is induced by the Fisher information matrix, whereas in our setting it is shaped by the path kernel.

### 4.1. Rescaling is pre-conditioning

If we replace a parameter $\theta$ (either at initialization or during training iterations) by a rescaled version $\theta' = D\theta$, $D \in \mathcal{D}$, then since $z = \Phi(\theta)$ is unchanged, only $P_\theta$ on the right-hand-side of (7) can depend on $D$. In other words, *selecting some admissible rescaling corresponds to shaping the geometry of the optimization dynamics in the lifted space.*

However, what criterion should guide the choice of a rescaling? Several heuristic approaches exist: Equinormalization (Stock et al., 2019; Saul, 2023) explicitly chooses $D$ to minimize some $\ell_{p,q}$ norm of $\theta' = D\theta$ or Mustafa & Burkholz (2024) select rescalings that promote more balanced weights across successive layers (for graph attention networks).

We show in Appendix C that on toy examples several natural criteria used in the literature are in fact equivalent: the Euclidean norms $\|\theta'\|_2$ and $\|\nabla L(\theta')\|_2$, the condition number $\kappa(\nabla^2 L(\theta'))$, the trace $\text{tr}(\nabla^2 L(\theta'))$, or the operator norm $\|\nabla^2 L(\theta')\|_{2\to2}$. However, this no longer holds for larger models, and an appropriate criterion must be selected.

### 4.2. Proposed rescaling criterion

Rather than the heuristic criteria to choose $D$ described above, `PathCond` relies on a criterion defined by the path kernel. Our strategy is based on the assumption that *gradient descent in the lifted space provides a suitable, rescaling-invariant idealized trajectory.* Accordingly, the rescaling $D$ is chosen to mimic this behavior by seeking to *best align* $P_{D\theta}\nabla\ell(\Phi(\theta)) \in \mathbb{R}^q$ with $\nabla\ell(\Phi(\theta)) \in \mathbb{R}^q$. A first naive approach would attempt to actually compute $\nabla\ell(\Phi(\theta))$ and optimize some correlation with $P_{D\theta}\nabla\ell(\Phi(\theta))$. This would however be highly impractical for practical networks as the dimension $q$ of these vectors is the number of *paths* in the graph associated to the network, which grows combinatorially with the network size (e.g., it is in the order of $q = 10^{53}$ for ResNet18, details in Appendix I).

Instead, we propose to directly seek an alignment between $P_\theta$ and the identity matrix. As before, a naive attempt to explicitly compute and optimize $P_\theta \in \mathbb{R}^{q \times q}$ is computa-

---

**Algorithm 1** Idealized Rescaling Algorithm

---

1: **Input:** Learning rates $(\mu_k)_{k \geq 0}$, loss function $L(\cdot)$, divergence $d(\cdot||\cdot)$.
2: Iterate following or only teleport initialization.
3: **for** $k = 0, 1, 2, \ldots$ **do**
4:     $(\alpha_k, D_k) = \underset{\alpha > 0, D \in \mathcal{D}}{\operatorname{argmin}}\ d(\alpha G_{D\theta_k}||I_p)$   // see Algorithm 2
5:     Set $\theta_{k+\frac{1}{2}} = D_k \theta_k$            // Teleportation step
6:     $\theta_{k+1} = \theta_{k+\frac{1}{2}} - \mu_k \nabla L(\theta_{k+\frac{1}{2}})$      // Gradient step
7: **end for**

---

tionally infeasible. However, as we show next, a carefully designed criterion enables us to entirely bypass this bottleneck: we identify optimal rescalings—according to the proposed criterion—*without ever forming or storing $P_\theta$*. To this end, we must address two challenges: *how to measure the "distance" between $P_\theta$ and the identity matrix $I$, and how to circumvent the computational bottleneck arising from the combinatorial size $q \times q$ of $P_\theta$.*

To address the first challenge, we use Bregman matrix divergences $d_\zeta(X||Y)$, induced by a strictly convex function $\zeta$ (Kulis et al., 2009), which provide a principled geometry on the space of symmetric matrices (see Appendix E for a formal definition). In particular, we focus on *spectral divergences*, a subclass defined via functions of the eigenvalues, encompassing common choices such as the Frobenius and logdet (James et al., 1961) divergences.

For the second question, we note that $P_\theta$ can be expressed as $P_\theta = AA^\top$, where $A = \partial\Phi(\theta)$. Using any divergence satisfying the property $d(AA^\top||I_q) = d(A^\top A||I_p)$ for every $q \times p$ matrix $A$, we will be able to work with the matrix $G_\theta := A^\top A = \partial\Phi(\theta)^\top \partial\Phi(\theta)$, which has size $p \times p$, with $p$ the number of parameters. In typical neural networks, we have $p \ll q$, making $G_\theta$ significantly more manageable[5] than $P_\theta$. As written in Appendix E, the above property is satisfied with spectral divergences, as $AA^\top$ and $A^\top A$ share the same spectrum.

**Generic conditioning algorithm.** Given such a spectral-based divergence measure $d(\cdot||\cdot)$, the above viewpoint combined with the teleportation-based algorithm of Zhao et al. (2022), leads to Algorithm 1. Note that we add a coefficient $\alpha_k$ in the optimization step to manage the relative scale of the kernel with respect to the identity matrix.

### 4.3. Explicit algorithm with the logdet divergence

The final step is to select an appropriate divergence. We choose the Bregman divergence induced by $\zeta(X) = -\log\det(X)$, defined on positive semi-definite matrices.

---

[5]With our final criterion, only the diagonal of $G_\theta$ is required.

In light of this, we focus on the optimization problem

$$\min_{D \in \mathcal{G},\ \alpha > 0} d_{-\log\det}(\alpha\, G_{D\theta} \,\|\, I_p)\,. \tag{8}$$

Interestingly, $d_{-\log\det}(X||I)$ provides an upper bound on the condition number of $X$ (Dhillon, 2008; Bock & Andersen, 2025), further supporting the interpretation of our approach as selecting a rescaling that improves the condition number of the path kernel.

To handle potential issues arising from zero eigenvalues, we consider in Appendix F a slight variation that employs the generalized determinant in place of the standard det. Furthermore, as shown in Appendix F, if $G := G_\theta$ is positive definite, that is when $\partial\Phi$ is full rank, then $G_{D\theta}$ remains positive definite for every $D \in \mathcal{D}$, and the resulting optimization problem coincides with the formulation using the standard $\log\det$. The latter can be rewritten as (Lemma F.3)

$$\min_{u \in \mathbb{R}^H}\ F(u) := p \log\left(\sum_{i=1}^{p} e^{(Bu)_i} G_{ii}\right) - \sum_{i=1}^{p} (Bu)_i\,, \tag{9}$$

where $H$ is the number of hidden neurons, and $B \in \mathbb{R}^{p \times H}$ is a matrix with entries in $\{1, -1, 0\}$, indicating whether a parameter enters $(-1)$, leaves $(1)$, or is unrelated to $(0)$ a given neuron. Each column of $B$ corresponding to a neuron $h \in H$ takes the form $b_h = (1_{\mathrm{out}_h}, -1_{\mathrm{in}_h}, 0_{\mathrm{other}_h})^\top \in \mathbb{R}^p$ (up to a permutation of coordinates), where $\mathrm{out}_h$, $\mathrm{in}_h$, and $\mathrm{other}_h$ are defined formally in Definition D.1. The rescaling of $h$ is given by $\lambda_h = e^{u_h}$ and $D = e^{Bu}$ (the exponential is taken entrywise) describes the overall rescaling on all parameters. As shown in Lemma F.6 in appendix, $F$ is convex and (9) admits a solution as soon as $g := \mathrm{diag}(G) > 0$. Although $G$ is generally only positive *semi*-definite ($g \geq 0$), we nonetheless adopt (9) as the basis for our concrete criterion.

Indeed, the reformulation (9) presents several practical advantages. First, the objective $F$ is defined in dimension $H$, corresponding to the number of neurons, which is typically much smaller than the number of parameters $p$. Moreover, $F$ depends only on $g = \mathrm{diag}(G)$ and therefore does not require the explicit computation of the full $p \times p$ matrix $G_\theta$. The vector $g$ can be itself computed efficiently using automatic differentiation using a single backward pass on the network (see Appendix G). Finally, (9) can be solved efficiently via alternating minimization as described below.

**Lemma 4.1.** *If $g > 0$, for any neuron $h$ the problem $\min_{u_h \in \mathbb{R}} F(u_1, \cdots, u_h, \cdots, u_H)$ has a solution given by $\log(r_h)$ where $r_h$ is the unique positive root of the polynomial $\mathcal{B}(\mathcal{A} + p)X^2 + \mathcal{A}\mathcal{D}X + \mathcal{C}(\mathcal{A} - p)$ where, with*

**Algorithm 2** `PathCond` : Path-conditioned rescaling

**input** : DAG ReLU network, $\theta$ to rescale

1   $u^{(0)} = 0 \in \mathbb{R}^H$

2   **for** $k = 0, \ldots, n_{iter}$ **do**

3     **for** *each neuron $h$* **do**

4       Given $u^{(k)}$ compute $\mathcal{A}, \mathcal{B}, \mathcal{C}, \mathcal{D}$ defined in Lemma 4.1.

5       Solve $r_h$ = `find_positive_root`$(\mathcal{B}(\mathcal{A} + p)X^2 + \mathcal{A}\mathcal{D}X + \mathcal{C}(\mathcal{A} - p)) > 0$.

6       $u_h^{(k+1)} = \log(r_h)$

**output** : Rescaled $D\theta$ where $D = \mathrm{diag}(e^{Bu^{(n_{\mathrm{iter}})}})$

---

$\rho_{i,h} := e^{(Bu)_i - B_{ih}u_h}$, *we consider*

$$\mathcal{A} := |\{i \in \mathrm{in}_h\}| - |\{i \in \mathrm{out}_h\}|, \ \mathcal{B} := \sum_{i \in \mathrm{out}_h} \rho_{i,h}g_i$$

$$\mathcal{C} := \sum_{i \in \mathrm{in}_h} \rho_{i,h}g_i, \ \mathcal{D} := \sum_{i \in \mathrm{other}_h} \rho_{i,h}g_i.$$

We provide a detailed proof of this lemma in Appendix F. Based on this result, we use alternating minimization with closed-form coordinate updates, yielding Algorithm 2.

**Computational complexity.** Algorithm 2 has time complexity $O(n_{\mathrm{iter}}p)$ and space complexity $O(p + H)$. In practice, $n_{\mathrm{iter}} \ll p$ with typical convergence in fewer than 10 iterations (ablation study in Appendix H.2). Indeed, the computation of $\mathrm{diag}(G)$ requires one backward pass at cost $O(p)$. We never compute $Bu$ explicitly: at the first iteration $Bu = u = 0$, and thereafter we incrementally update $Bu$ by adding the coefficient difference at cost $|\mathrm{out}_h| + |\mathrm{in}_h|$ per iteration. Each iteration cycles through $H$ neurons, where updating neuron $h$ costs $O(|\mathrm{out}_h| + |\mathrm{in}_h|)$. Since $\sum_{h=1}^{H}(|\mathrm{out}_h| + |\mathrm{in}_h|) \approx 2p$, each iteration has complexity $O(p)$. The algorithm stores dual variables $Bu \in \mathbb{R}^p$, primal variables $u \in \mathbb{R}^H$, and a sparse matrix $B$ with $\approx 2p$ nonzero coefficients (all $\pm 1$) and index tensors for edge sets, yielding space complexity $O(p + H)$. In Section 5.3, we further show that `PathCond` runs in at most the time of a single training epoch across several architectures on CIFAR-10. Full implementation details, pseudocode, and timing measurements are provided in Appendix H.

## 5. Experiments

In this section, we demonstrate that rescaling *only at initialization* with `PathCond` improves training dynamics by accelerating training loss convergence. In a nutshell `PathCond` enables us to reach the same accuracy with up to 1.5 times fewer epochs without compromising generalization (Figure 3). We also compare our approach with the baseline GD with standard initialization without any rescaling, and the Equinormalization heuristic (Stock et al., 2019)

(`ENorm` ), which performs rescaling at every iteration to minimize a weighted $\ell_{2,2}$ norm of the layer weights. Unlike the `ENorm` criterion and its variants (Saul, 2023), which require tuning of hyper parameters (the choice of p,q and of the weights), the `PathCond` criterion has no hyperparameter. Full details of the `ENorm` configuration used in our experiments are provided in Appendix N. Finally, we characterize theoretically the favorable regimes for `PathCond` and validate them on MNIST autoencoders (Figure 5).

### 5.1. Faster Training Dynamics

**Training setup.** Following the experimental protocol of Stock et al. (2019), we evaluate our method on fully connected networks for CIFAR-10 classification. The models take flattened 3072-dimensional images as input. The architecture consists of an initial linear layer ($3072 \times 500$), followed by $L - 1$ hidden layers of size $500 \times 500$, and a final classification layer ($500 \times 10$), with BatchNorm applied after each hidden linear layer. We vary the depth $L$ from 3 to 9 to analyze depth-dependent effects. All weights are initialized using Kaiming uniform initialization (He et al., 2015). We do 100 epochs of SGD, with a batch size of 128 and a fixed learning rate (for all architectures) of $10^{-3}$. Results are averaged over 3 independent runs.

**Results.** The results are shown in Figure 2, where we report the number of parameters required to reach $99\%$ training accuracy (left plot). Overall, `PathCond` reaches target accuracy up to $1.5\times$ faster at the same learning rate, and as a side effect also improves parameter efficiency: a 2.5M parameter model matches the performance of 3.25M parameter baseline models. The improvement magnitude varies with the depth. The 3-hidden-layer network (2M parameters) exhibits the strongest gains: training accuracy converges faster with `PathCond` initialization and the training loss decreases more rapidly toward zero. Deeper networks show more modest but consistent advantages. Notably, `ENorm` does not yield improvements in this experiment, whereas only `PathCond` achieves speedups.

Additional learning rates are reported in Appendix K. `PathCond` matches or exceeds baseline performance across small to moderate learning rates, with the greatest improvements observed at small values where initialization most critically influences training dynamics. As the learning rate increases, the effect becomes less pronounced, yet remains comparable to the baseline and `ENorm` .

### 5.2. Generalization

While our method is designed to improve training loss convergence, it does not explicitly target generalization. In this experiment, we show that `PathCond` accelerates training without degrading generalization performance.

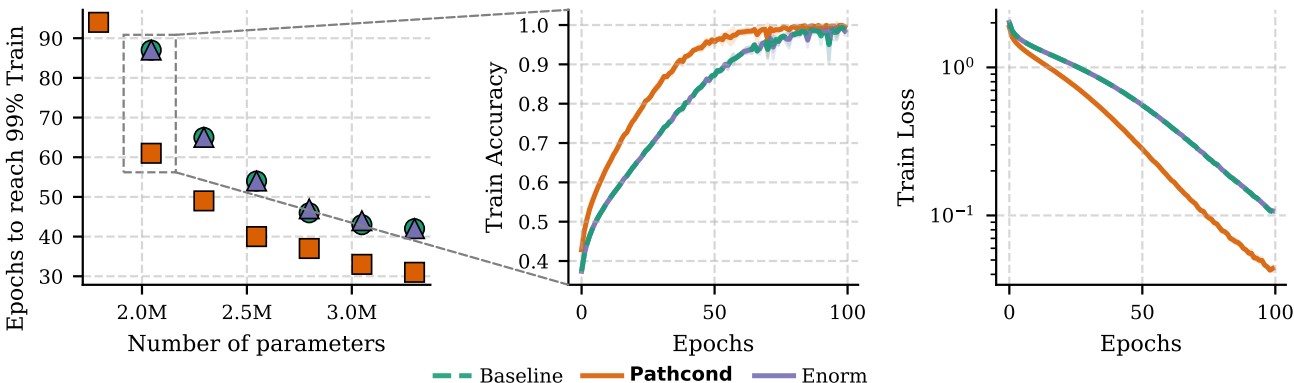

*Figure 2.* PathCond performance comparison across network depths on CIFAR-10 with multilayer perceptrons (**Left**) Number of epochs required to reach 99% training accuracy for networks with 2 to 8 hidden layers (abscissa = number of parameters, which increases with depth). (**Middle**) Training accuracy curves for the 3-hidden-layer network. (**Right**) Corresponding training loss curves

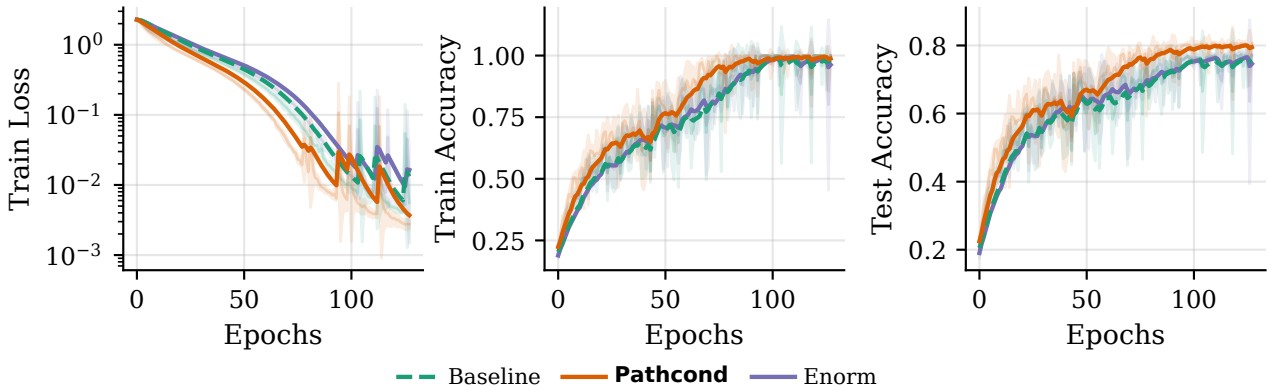

*Figure 3.* Training dynamics on CIFAR-10 with a fully convolutional architecture (CIFAR-NV). (**Left**) Training loss, (**Middle**) training accuracy, and (**Right**) test accuracy. Curves are smoothed using an exponential moving average with factor $\alpha = 0.2$.

**Training Setup.** We evaluate our method on the CIFAR-NV fully convolutional architecture (Gitman & Ginsburg, 2017). The architecture processes CIFAR-10 images with channel-wise normalization. The original training set is split into 40,000 images for training and 10,000 for validation. Training is performed for 128 epochs using SGD with a learning rate of 0.001. Weights are initialized using Kaiming's initialization scheme. We employ a BatchNorm layer after the final fully connected layer for all methods (Baseline, `ENorm`, and `PathCond`). Additional learning rates are reported in Appendix L.

**Results.** Figure 3 shows the training dynamics with the fully convolutional architecture. `PathCond` achieves faster convergence in both training loss (left panel) and training accuracy (middle panel), reaching near-optimal performance approximately 20 epochs earlier than baseline methods. Critically, this acceleration translates to improved test accuracy (right panel): `PathCond` reaches 80% best test accuracy while Baseline and `ENorm` plateau at 77%, demonstrating that the improved training dynamics could also enhance

rather than compromise generalization.

### 5.3. Computational Cost

A key advantage of `PathCond` is its computational efficiency. As detailed in Section 4.3, `PathCond` does not require computing any full Jacobian or matrix decomposition. Using the logdet divergence, we build a criterion that only depends on the diagonal elements of the matrix $G = \partial\Phi^\top\partial\Phi$, which costs $O(\# \text{params})$ to compute and can be calculated with a single backward pass. Our algorithm to solve the resulting optimization problem using block coordinate descent is also efficient.

**Experimental protocol.** We measure wall-clock times on an NVIDIA RTX A4000 GPU across the architectures used throughout the paper: the MLPs with BatchNorm described in Section 5.1 (depth $L = 4$ with hidden width 500; additional depths in Appendix H.1), the fully convolutional CIFAR-NV network from Section 5.2, and ResNet-18 of type C (He et al., 2016), in which all shortcuts are learned

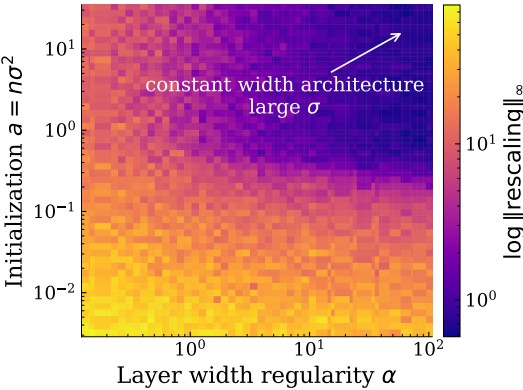

*Figure 4.* Analysis of the relationship between layer width regularity (controlled by the ratio $\alpha$) and log-rescaling magnitude for small and large variance regimes.

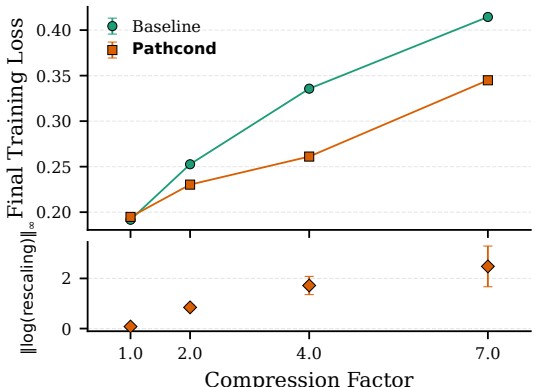

*Figure 5.* Effect of compression on MNIST autoencoder training. (**Top**) Final training loss for different compression factors. (**Bottom**) Maximum absolute value of the log rescaling at initialization.

$1 \times 1$ convolutions. For each architecture, we compare: (1) the time to perform one epoch of SGD training on CIFAR-10 (batch size 128), and (2) the time to compute the full PathCond rescaling at initialization. The rescaling consists of computing the diagonal of the matrix $G$ followed by 10 iterations of block coordinate descent to solve the optimization problem. Training and rescaling times are averaged over 10 runs (after a warm-up phase to account for JIT compilation). We report means and standard deviations.

**Results.** As shown in Figure 6, the rescaling cost is at most comparable to a single training epoch across all architectures tested, ranging from 22% (CIFAR-NV) to 103% (MLP with 3 hidden layers) of one epoch time. Since, in practice, rescaling is performed *once* at initialization, the overhead over a full training run (typically 100+ epochs) is negligible. Further optimizations appear feasible by parallelizing block coordinate descent along even/odd layers, which we leave for future work.

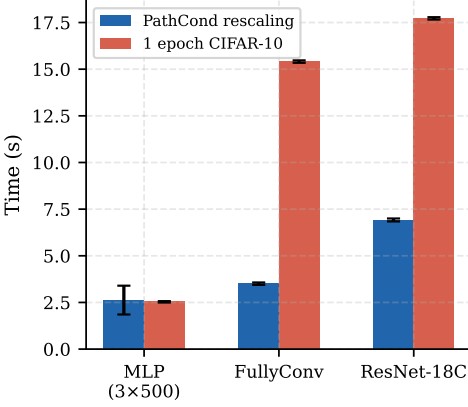

*Figure 6.* Wall-clock time comparison on an NVIDIA RTX A4000. PathCond rescaling time versus one epoch of CIFAR-10 training across different architectures.

### 5.4. What are the Favorable Regimes for `PathCond` ?

As observed in Figure 2, network architecture influences the effectiveness of our algorithm. This raises a natural question: are there architectures or initializations for which $G$ is already close to the identity (according to the Bregman divergence) without rescaling $\theta$ ? Understanding such cases helps identify when `PathCond` provides the greatest benefits. A simple example is when the diagonal of $G$ is constant, i.e., where $\exists \alpha > 0$ such that $g = \alpha \mathbf{1}$. In this case, the objective function (9) admits $u = 0$ as its unique minimizer. Consequently, the rescaling matrix is the identity and the algorithm has no effect (proof in Appendix J).

**Identifying favorable regimes.** Our goal is to identify regimes in which the diagonal exhibits a *wide spread*, i.e., deviates significantly from being constant, by analyzing how $g$ depends on the network architecture and initialization. We hypothesize—and verify empirically—that significant spread in the diagonal correlates with the effectiveness of our rescaling algorithm.

For layered fully-connected ReLU networks (LFCNs) without biases[6], the expected diagonal coefficients at initialization can be computed explicitly from the network widths $n_0, \ldots, n_L$ and the variance of the initialization of each layer $\sigma_k^2$. Standard initializations such as Kaiming (He et al., 2015) have the property that $n_k \sigma_k^2$ is constant across all layers. We denote this value by $a := n_k \sigma_k^2$. We have:

**Proposition 5.1** (Expected diagonal under standard initialization). *For any edge parameter $i$ at layer $k \in \{0, \ldots, L-1\}$, the expected $i$th coefficient of $\mathrm{diag}(G)$ is given by*

$$\mathbb{E}[G_{ii}] = \frac{n_L}{n_{k+1}} a^{L-1} + \frac{n_L}{n_{k+1}} \sum_{j=0}^{k-1} \frac{a^{L-1-j}}{n_j}. \tag{10}$$

---
[6]biases are dealt with in Appendix J.2

The full derivation, including the case with biases, is provided in Appendix J. This result helps us identify regimes in which $\text{diag}(G)$ is close to constant. Consider networks with constant width across layers $n_0 = \ldots = n_L := n$:

- **Case** $a = 1$: (10) simplifies to $\mathbb{E}[G_{ii}] = 1 + \frac{k}{n}$. When $n \gg L$, we have $\frac{k}{n} \ll 1$ and thus $\mathbb{E}[G_{ii}] \approx 1$.

- **Case** $a \to +\infty$: We have $\mathbb{E}[G_{ii}] \sim a^{L-1}(1 + 1/n)$ for $1 \leq k \leq L - 1$. Again, for $n$ large enough, $\text{diag}(G)$ is close to $\alpha \mathbf{1}$ with $\alpha = a^{L-1}$.

In contrast, when $a \to 0$, for all weights on layer $k$, we have $\mathbb{E}[G_{ii}] \sim \frac{1}{n} a^{L-k}$. Because $a$ is small, there is a large difference in the expected values of $g$ for coefficients on different layers, creating significant spread. In Appendix J.4, we analyze networks with non-constant widths and show that $g$ depends on the width ratios between layers. Based on this analysis, we identify the two most favorable regimes:

1. **Varying width architectures:** When layer widths vary significantly ($n_i \neq n_j$), the diagonal exhibits a widespread regardless of initialization scale $a$.

2. **Small variance initialization:** As shown above, for near constant-width architectures ($n_i \approx n$ for all $i$), the diagonal exhibits significant spread when the initialization standard deviation $\sigma_k$ is small relative to $1/\sqrt{n}$, i.e., when $a \ll 1$.

In both cases, the diagonal magnitudes vary widely across parameters, creating favorable conditions for PathCond .

**Empirical validation.** To validate these predictions, we generate networks of fixed depth (8 layers) and fixed mean width (32 neurons per layer), varying only the *architectural regularity* i.e., how uniformly neurons are distributed across layers. This is controlled by a Dirichlet-based sampling scheme (detailed in Appendix J) with concentration parameter $\alpha$ that interpolates between highly non-uniform layer widths ($\alpha \to 0$) and perfectly uniform ones ($\alpha \to +\infty$, all layers with $\approx 32$ neurons). For each sampled architecture, we initialize the weights at a range of variances and compute $\|\log(\text{rescaling})\|_\infty$, which quantifies the magnitude of the rescaling adjustments required by PathCond . Figure 4 reveals two clear trends, both consistent with our theoretical predictions: the rescaling magnitude grows as layer widths become more irregular, and as the initialization variance decreases.

**Experiment on the MNIST Autoencoder.** To further validate our characterization of favorable regimes, we consider an autoencoder task where we can systematically vary architectural balance through the compression factor. We follow the experimental setup of Desjardins et al. (2015).

The encoder is a fully connected ReLU network with architecture $[784, 784/c_f, 784/c_f^2]$ where the compression factor $c_f \in \{1, 2, 4, 7\}$ (the decoder is symmetric). All weights are initialized using Kaiming uniform initialization. We train for 500 epochs using SGD with a batch size of 128 and learning rate $10^{-3}$ (additional results in Appendix M). Each experiment is repeated over 3 independent runs.

**Results.** Figure 5 shows the final training loss as a function of the compression factor. The effect of PathCond is more pronounced for larger compression factors, which produce more unbalanced architectures, consistent with our theoretical predictions. In all cases, PathCond achieves smaller training loss at convergence.

# 6. Conclusion and discussions

PathCond is a rescaling strategy based on geometric principles and on the path-lifting that aims to accelerate the training of ReLU networks. Our criterion is motivated by the idea that promoting a better-conditioned gradient flow in the network's lifted space provides a principled approach for minimizing the training loss. This work opens several promising directions. A natural one would be to adapt the method for adaptive optimizers with momentum such as Adam (Kingma & Ba, 2014), Muon (Jordan et al., 2024) or Shampoo (Gupta et al., 2018). These methods are complementary to our approach and combining them with the rescaled initialization of PathCond would be an interesting step for future work. Another direction would be to investigate applications to modern architectures including transformers with self-attention mechanisms.

Moreover, while PathCond has been designed for positively 1-homogeneous activation functions (in order to have the rescaling property), we show empirically in Appendix O that the rescaling remains approximately input-output preserving for smooth activations such as GELU and SiLU, and still accelerates training in this regime. This robustness opens the door to applying PathCond to modern architectures that rely on smooth, non-homogeneous activations—most notably transformers, whose feed-forward blocks now commonly use GELU or gated variants such as SwiGLU (Shazeer, 2020). Extending both the analysis and the rescaling procedure to such gated activations constitutes a promising direction for future research.

## Impact Statement

This paper presents work whose goal is to advance the field of machine learning. There are many potential societal consequences of our work, none of which we feel must be specifically highlighted here.

## Acknowledgements

This project was supported by the SHARP project of the PEPR-IA (ANR-23-PEIA-0008, granted by France 2030). We thank the Blaise Pascal Center for its computational support, using the SIDUS solution (Quemener & Corvellec, 2013). All experiments were implemented in Python using PyTorch (Paszke et al., 2019) and NumPy (Harris et al., 2020). We thank Manon Verbockhaven for insightful discussions throughout the course of this work.

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

## A. Properties of the path-lifting

We have the following properties on the path-lifting

**Lemma A.1.** *Let $D \in \mathcal{D}, \partial\Phi(D\theta) = \partial\Phi(\theta)D^{-1}$.*

*Proof.* Using the definition of the path-lifting, $\Phi(\theta) := (\Phi_p(\theta))_p$ with $\Phi_p(\theta) := \prod_{i \in p} \theta_i$, we have

$$\frac{\partial\Phi_p(\theta)}{\partial\theta_i} = \begin{cases} \prod_{\substack{k \in p \\ k \neq i}} \theta_k & \text{if } p \ni i \\ 0 & \text{otherwise}. \end{cases} \tag{11}$$

Given any $D \in \mathcal{D}$, we have for all $\theta \in \mathbb{R}^p$, $\Phi(D\theta) = \Phi(\theta)$ (Gonon, 2024, Theorem 2.4.1). This means the functions $\theta \mapsto \Phi(\theta)$ and $\theta \mapsto \Phi(D\theta)$ are identical and they share the same Jacobian. Applying the chain rule yields

$$\partial\Phi(\theta) = \partial\Phi(D\theta)D \tag{12}$$

The conclusion follows by multiplying on the right by $D^{-1}$, which exists by Lemma D.3. $\qquad\square$

## B. Gradient in $\Phi$-space

As described in the main text, if $\theta(t)$ follows the gradient flow $\dot\theta = -\nabla L(\theta)$ of (1) then $z(t) := \Phi(\theta(t))$ satisfies $\dot z = \partial_t \Phi(\theta) = -P_\theta \nabla_\Phi \ell(\Phi(\theta))$ with $P_\theta = \partial\Phi(\theta)\partial\Phi(\theta)^\top$. As an alternative to the proposed `PathCond` approach to "align" $P_\theta \nabla_\Phi \ell(\Phi(\theta))$ with $\nabla_\Phi \ell(\Phi(\theta))$, one could envision a "natural gradient" version along the following route (Amari, 1998; Verbockhaven et al., 2024). Considering any $p \times p$ pre-conditioning matrix $M(\theta)$, it is possible to replace (1) with

$$\dot\theta = -M(\theta)\nabla L(\theta). \tag{13}$$

With the same reasoning based on chain rules, if the trajectory $\theta(t)$ is governed by such an ODE then $z(t)$ satisfies the modified ODE

$$\dot z = -\partial\Phi(\theta)\dot\theta = -\partial\Phi(\theta)M(\theta)\nabla L(\theta) = -\partial\Phi(\theta)M(\theta)\partial\Phi(\theta)^\top \nabla_\Phi \ell(\Phi(\theta)). \tag{14}$$

The best alignment with $\nabla_\Phi \ell(\Phi(\theta))$ is achieved with the pseudo-inverse $M(\theta) := [\partial\Phi(\theta)^\top \partial\Phi(\theta)]^+$, leading to

$$\dot z = -Q_\theta \nabla_\Phi \ell(\Phi(\theta)). \tag{15}$$

with $Q_\theta$ the $q \times q$ matrix projecting orthogonally onto $\text{range}(\partial\Phi(\theta))$. While this approach leads to the maximum alignment, it involves the computation of the pseudo-inverse of a large $p \times p$ matrix, which is a strong computational bottleneck that `PathCond` circumvents.

## C. Equivalence between different rescaling criteria on toy examples

**First example.** Consider a minimal architecture with two weights $\theta = (u, v)$, $f_\theta(x) := uvx$, and a loss $L(\theta) = \ell(\Phi(\theta))$ where $\Phi(\theta) := uv$. Straightforward calculus yields

$$\nabla\Phi(\theta) = \begin{pmatrix} v \\ u \end{pmatrix} = \begin{pmatrix} 0 & 1 \\ 1 & 0 \end{pmatrix} \theta \quad \text{and } \nabla^2\Phi(\theta) = \begin{pmatrix} 0 & 1 \\ 1 & 0 \end{pmatrix}$$

$$\nabla L(\theta) = \ell'(\Phi(\theta)) \cdot \nabla\Phi(\theta)$$

$$\nabla^2 L(\theta) = \underbrace{\ell''(\Phi(\theta))}_{=:a} \cdot \nabla\Phi(\theta)(\nabla\Phi(\theta))^\top + \underbrace{\ell'(\Phi(\theta))}_{=:b} \cdot \nabla^2\Phi(\theta)$$

$$= a \begin{pmatrix} v^2 & uv \\ uv & u^2 \end{pmatrix} + b \begin{pmatrix} 0 & 1 \\ 1 & 0 \end{pmatrix} = \begin{pmatrix} av^2 & a\Phi(\theta) + b \\ a\Phi(\theta) + b & au^2 \end{pmatrix}$$

Since $\Phi$ is rescaling invariant it follows that for any norm $\|\cdot\|$

$$\arg\min_{\theta' \sim \theta} \|\nabla L(\theta)\| = \arg\min_{\theta' \sim \theta} \|\nabla\Phi(\theta)\| = \arg\min_{\theta' \sim \theta} \|\theta\| \tag{16}$$

and for the special case of the Euclidean norm

$$\arg\min_{\theta'\sim\theta} \operatorname{tr}(\nabla^2 L(\theta)) = \arg\min_{\theta'\sim\theta} \|\theta\|_2. \tag{17}$$

We now turn to the behavior of the condition number $\kappa(\nabla^2 L(\theta))$ and the operator norm $\|\nabla^2 L(\theta)\|_{2\to2}$ by considering the spectrum of $\nabla^2 L(\theta)$, which is made of two eigenvalues $\lambda_- \leq \lambda_+$ such that

$$
\begin{aligned}
\lambda_+\lambda_- = \det(\nabla^2 L(\theta)) &= a^2(uv)^2 - (a\Phi(\theta)+b)^2 \\
&= (a\Phi(\theta))^2 - (a\Phi(\theta)+b)^2 =: -b(2a\Phi(\theta)+b) =: c \\
\lambda_+ + \lambda_- = \operatorname{tr}(\nabla^2 L(\theta)) &= a(u^2+v^2) = a\|\theta\|_2^2.
\end{aligned}
$$

Assuming that $a \geq 0$ (this is the case for example when $\ell$ is the quadratic loss), the sum $\lambda_+ + \lambda_-$ is thus non-negative, hence $\lambda_+ \geq |\lambda_-|$ and therefore $\|\nabla^2 L(\theta)\|_{2\to2} = \lambda_+$. We also obtain that $\kappa(\nabla^2 L(\theta)) = \lambda^+/|\lambda_-|$. Notice that each of these quantities can be expressed as $g(\lambda_+, |\lambda_-|)$ where $g$ increases with its first argument, and decreases with is second one. We now investigate how they vary with $\|\theta'\|_2^2$ when $\theta' \sim \theta$.

Since $a, b, c$ can be expressed as a function of $\Phi(\theta)$, they are unchanged if we replace $\theta$ by a rescaling equivalent paramter $\theta' \sim \theta$. When varying such a $\theta'$, an increase in $\|\theta'\|_2^2$ increases the sum $\lambda_+ + \lambda_-$ while leaving the product $\lambda_+\lambda_-$ constant. Let us distinguish two cases:

- case $c \geq 0$: both eigenvalues have the same sign, and since their sum $a\|\theta'\|_2^2$ is positive, they are both positive. Straightforward calculus shows that increasing their sum while keeping their product constant increases the largest one $\lambda_+$ while decreasing the smallest one $\lambda_- = |\lambda_-|$. Therefore with $g$ as above the value $g(\lambda_+, \lambda_-)$ increases with $\|\theta'\|_2^2$

- case $c < 0$ here since we have seen that $\lambda_+$ remains positive we must have $\lambda_- = -|\lambda_-|$. The same type of reasoning shows that we reach the same conclusion.

Overall it follows that

$$\arg\min_{\theta'\sim\theta} \kappa(\nabla^2 L(\theta)) = \arg\min_{\theta'\sim\theta} \|\nabla^2 L(\theta)\|_{2\to2} = \arg\min_{\theta'\sim\theta} \|\theta\|_2. \tag{18}$$

**Second example.** The above example is a particular case of a deeper toy example where $\theta = (u_1, \ldots, u_L)$ and $\Phi(\theta) = u_1 \ldots u_L$. Similar computations (we skip the details) show that when all entries $u_i$ are nonzero, denoting $\theta^{-1} := (u_i^{-1})_{i=1}^L$

$$
\begin{aligned}
\nabla\Phi(\theta) &= (\Phi(\theta)/u_i)_{i=1}^L = \Phi(\theta) \cdot \theta^{-1} \\
\nabla L(\theta) &= \ell'(\Phi(\theta)) \cdot \nabla\Phi(\theta) = \underbrace{\ell'(\Phi(\theta)) \cdot \Phi(\theta)}_{=:a(\Phi(\theta))} \cdot \theta^{-1} \\
\nabla^2 L(\theta) &= a(\Phi(\theta)) \cdot \operatorname{offdiag}(\theta^{-1}(\theta^{-1})^\top) + (\Phi(\theta))^2 \ell''(\Phi(\theta)) \cdot \theta^{-1}(\theta^{-1})^\top
\end{aligned}
$$

We deduce that $\operatorname{tr}(\nabla^2 L(\theta)) = (\Phi(\theta))^2 \ell''(\Phi(\theta)) \|\theta^{-1}\|_2^2$. Overall we obtain with the same reasoning that

$$\arg\min_{\theta'\sim\theta} \|\nabla L(\theta)\| = \arg\min_{\theta'\sim\theta} \operatorname{tr}(\nabla^2 L(\theta)) = \arg\min_{\theta'\sim\theta} \|\theta^{-1}\|_2. \tag{19}$$

## D. Characterization of the rescaling symmetry

Let $\mathcal{G} = (V, E)$ be a fixed directed acyclic graph (DAG), representing a fixed network architecture, with vertices $V$ called neurons and edges $E$. For a neuron $v \in V$, we define the sets of antecedents and successors as

$$\operatorname{ant}(v) := \{u \in V \mid (u, v) \in E\}, \quad \operatorname{suc}(v) := \{u \in V \mid (v, u) \in E\}. \tag{20}$$

Neurons with no antecedents (resp. no successors) are called *input* (resp. *output*) neurons, and their sets are denoted $N_{\text{in}} := \{v \in V \mid \operatorname{ant}(v) = \emptyset\}$ and $N_{\text{out}} := \{v \in V \mid \operatorname{suc}(v) = \emptyset\}$ respectively. The input and output dimensions are $d_{\text{in}} := |N_{\text{in}}|$ and $d_{\text{out}} := |N_{\text{out}}|$. Neurons in $\mathcal{H} := V \setminus (N_{\text{in}} \cup N_{\text{out}})$ are called *hidden neurons*, and we denote their cardinality

by $H := |\mathcal{H}|$. Note that for any hidden neuron $v \in \mathcal{H}$, we have $\text{ant}(v) \neq \emptyset$ and $\text{suc}(v) \neq \emptyset$, i.e., there exist both incoming and outgoing edges (Gonon, 2024, Definition 2.2.2).

To parametrize this graph in the context of neural networks, we perform a topological ordering (Cormen et al., 2022) and assign indices to all neurons and edges. All parameters (weights along edges and biases on neurons) can then be arranged into a vector $\boldsymbol{\theta} \in \mathbb{R}^p$, where $p$ is the total number of parameters. As the architecture is fixed, this vector fully represents our network.

**Definition D.1** (Neuron-wise rescaling symmetry). Let $v \in \mathcal{H}$ be a hidden neuron and $\lambda > 0$ a scaling factor. We define the sets of incoming and outgoing parameter indices associated to $v$ as

$$\text{in}_v := \{\text{index}(b_v)\} \cup \{\text{index}(\theta_{u \to v}) \mid u \in \text{ant}(v)\} \tag{21}$$

and

$$\text{out}_v := \{\text{index}(\theta_{v \to u}) \mid u \in \text{suc}(v)\}, \tag{22}$$

where $b_v$ denotes the bias of neuron $v$, and $\theta_{u \to v}$ (resp. $\theta_{v \to u}$) denotes the weight of edge $(u, v) \in E$ (resp. $(v, u) \in E$) (Gonon, 2024, Definition 2.2.2). The diagonal matrix $D_{\lambda, v} \in \mathbb{R}^{p \times p}$ associated to the *neuron-wise rescaling symmetry* (Stock & Gribonval, 2023, Def. 2.3) is defined by its diagonal entries:

$$(D_{\lambda, v})_{ii} = \begin{cases} \lambda & \text{if } i \in \text{in}_v \\ 1/\lambda & \text{if } i \in \text{out}_v \\ 1 & \text{otherwise.} \end{cases} \tag{23}$$

This matrix implements a rescaling of neuron $v$ by factor $\lambda$: incoming weights and bias are multiplied by $\lambda$, while outgoing weights are divided by $\lambda$.

We can then define the following set:

**Definition D.2** (Rescaling symmetry group). We denote by $\mathcal{D}$ the subgroup generated by the neuron-wise rescaling symmetries (Stock & Gribonval, 2023, Def. 2.3):

$$\mathcal{D} = \text{gr}\left(\{D_{\lambda, v} \mid v \in \mathcal{H}, \lambda \in \mathbb{R}_*^+\}\right). \tag{24}$$

**Lemma D.3.** *The rescaling symmetry group admits the following explicit characterization:*

$$\mathcal{D} = \left\{ \prod_{v \in \mathcal{H}} D_{\lambda_v, v} \;\middle|\; \boldsymbol{\lambda} \in (\mathbb{R}_*^+)^H \right\}. \tag{25}$$

*The order of the product does not matter since diagonal matrices commute.*

*Proof.* Let $A := \{\prod_{v \in \mathcal{H}} D_{\lambda, v} \mid \boldsymbol{\lambda} \in (\mathbb{R}_*^+)^H\}$. Since diagonal matrices commute and $D_{\lambda, v}^{-1} = D_{1/\lambda, v}$, it is straightforward to show that $A$ is stable under product and inversion, and is therefore a subgroup of the group of strictly positive diagonal matrices. Let us show that it contains all $D_{\lambda, v}$. Indeed, given any $v \in \mathcal{H}$ and $\lambda > 0$, choosing $\lambda_v = \lambda$ and $\lambda_{v'} = 1$ for all $v' \in \mathcal{H} \backslash \{v\}$ yields $\prod_{v' \in \mathcal{H}} D_{\lambda_{v'}, v'} = D_{\lambda, v}$. By minimality of the generated subgroup, we therefore have $\mathcal{D} \subset A$. Conversely, every element of $A$ is a product of elements of $\mathcal{D}$, hence an element of $\mathcal{D}$ by the group property. $\qquad \square$

For the next definitions, we recall some notation from Gonon et al. (2023).

**Definition D.4** (Path in a DAG). (Gonon et al., 2023, Definition A.1) A path of a DAG $\mathcal{G} = (V, E)$ is any sequence of neurons $v_0, \ldots, v_d$ such that each $v_i \to v_{i+1}$ is an edge . Such a path is denoted $\text{p} = v_0 \to \ldots \to v_d$. The length of a path $\text{p} = v_0 \to \ldots \to v_d$ is $\text{length}(\text{p}) = d$ (the number of edges).

**Definition D.5** (Path lifting). (Gonon et al., 2023, Definition A.2) For a path $\text{p} \in P$ with $\text{p} = v_0 \to \ldots \to v_d$, the path lifting is defined as

$$\Phi_{\text{p}}(\boldsymbol{\theta}) := \begin{cases} \displaystyle\prod_{\ell=1}^{\text{length}(\text{p})} \theta_{v_{\ell-1} \to v_\ell} & \text{if } v_0 \in N_{\text{in}}, \\ b_{v_0} \displaystyle\prod_{\ell=1}^{\text{length}(\text{p})} \theta_{v_{\ell-1} \to v_\ell} & \text{otherwise,} \end{cases} \tag{26}$$

where an empty product equals 1 by convention. The path-lifting $\Phi(\boldsymbol{\theta})$ of $\boldsymbol{\theta}$ over the entire graph is

$$\Phi(\boldsymbol{\theta}) := \left(\Phi_{\mathrm{p}}(\boldsymbol{\theta})\right)_{\mathrm{p}\in\mathrm{P}}. \tag{27}$$

By a slight abuse of notation, we extend the path-lifting to diagonal matrices by applying it to their diagonal entries. For $D \in \mathbb{R}^{p\times p}$ diagonal, we define

$$\Phi(D) := \Phi\left((D_{ii})_{i=1,\ldots,p}\right). \tag{28}$$

**Definition D.6** (Admissible rescaling matrices). We denote by $\Phi^{-1}(\mathbf{1})^+$ the set of diagonal matrices with strictly positive entries whose path-lifting equals the all-ones vector:

$$\Phi^{-1}(\mathbf{1})^+ := \left\{ D \in \mathbb{R}^{p\times p} \text{ diagonal} \,\big|\, \Phi(D) = \mathbf{1} \text{ and } D_{ii} > 0 \text{ for all } i \right\}. \tag{29}$$

**Lemma D.7.** *The set of admissible rescaling matrices coincides with $\Phi^{-1}(\mathbf{1})^+$:*

$$\mathcal{D} = \Phi^{-1}(\mathbf{1})^+. \tag{30}$$

*Proof.* We prove both inclusions.

- $\mathcal{D} \subseteq \Phi^{-1}(\mathbf{1})^+$

  Let $D \in \mathcal{D}$. By Lemma D.3, there exists $\boldsymbol{\lambda} \in (\mathbb{R}_*^+)^H$ such that

  $$D = \prod_{v\in\mathcal{H}} D_{\lambda_v,v}. \tag{31}$$

  Since $\boldsymbol{\lambda} \in (\mathbb{R}_*^+)^H$, we have $D_{ii} > 0$ for all $i$.

  It remains to show that $\Phi(D) = \mathbf{1}$. By Gonon (2024, Theorem 2.4.1), the rescaling symmetry property ensures that for any $\boldsymbol{\theta} \in \mathbb{R}^p$,

  $$\Phi(D\boldsymbol{\theta}) = \Phi(\boldsymbol{\theta}), \tag{32}$$

  Applying this property with $\boldsymbol{\theta} = \mathbf{1}$, we obtain

  $$\Phi(D\mathbf{1}) = \Phi(\mathbf{1}) = \mathbf{1}, \tag{33}$$

  where the last equality holds because $\Phi_{\mathrm{p}}(\mathbf{1}) = \prod_{i\in\mathrm{p}} 1 = 1$ for all paths $\mathrm{p} \in \mathrm{P}$.

  Now, observe that $(D\mathbf{1})_i = D_{ii}$ for all $i$. Therefore,

  $$\Phi(D) = \Phi(D\mathbf{1}) = \mathbf{1}. \tag{34}$$

  This shows that $D \in \Phi^{-1}(\mathbf{1})^+$, hence $\mathcal{D} \subseteq \Phi^{-1}(\mathbf{1})^+$.

- $\Phi^{-1}(\mathbf{1})^+ \subseteq \mathcal{D}$

  Let $D \in \Phi^{-1}(\mathbf{1})^+$. We can write $D = \mathrm{diag}(\boldsymbol{d})$ for some $\boldsymbol{d} \in (\mathbb{R}_*^+)^p$.

  We want to show that $D \in \mathcal{D}$, which by Lemma D.3 means showing that $D$ can be written as a product of neuron-wise rescaling matrices:

  $$D = \prod_{v\in\mathcal{H}} D_{\lambda_v,v} \tag{35}$$

  for some $\boldsymbol{\lambda} \in (\mathbb{R}_*^+)^H$.

  **Notation.** To simplify the exposition, for any neuron $v \in V$ and edge $(u,v) \in E$, we denote:

  $$d_v := d_{\mathrm{index}(b_v)} \quad\text{and}\quad d_{u\to v} := d_{\mathrm{index}(\theta_{u\to v})}, \tag{36}$$

  i.e., $d_v$ is the rescaling factor of the bias of neuron $v$, and $d_{u\to v}$ is the rescaling factor of the weight on edge $(u,v)$.

  **Characterization of $\mathcal{D}$.** Recall from Definition D.1 that the matrix $D_{\lambda_v,v}$ multiplies the bias and all incoming weights of neuron $v$ by $\lambda_v$, while dividing all outgoing weights by $\lambda_v$. Therefore, a diagonal matrix $D$ belongs to $\mathcal{D}$ if and only if there exist rescaling factors $\lambda_v > 0$ for each $v \in \mathcal{H}$ such that:

(i) For each hidden neuron $v \in \mathcal{H}$, the bias rescaling factor satisfies:

$$d_v = \lambda_v. \tag{37}$$

(ii) For each edge $(u, v) \in E$, the weight rescaling factor satisfies:

$$d_{u \to v} = \frac{\lambda_v}{\lambda_u}, \tag{38}$$

where we adopt the convention $\lambda_u = 1$ for $u \in N_{\text{in}}$ and $\lambda_v = 1$ for $v \in N_{\text{out}}$.

Indeed, condition (i) ensures that each bias is rescaled by the appropriate factor, while condition (ii) captures the combined effect on edges: as an outgoing edge from $u$, the weight is divided by $\lambda_u$; as an incoming edge to $v$, it is multiplied by $\lambda_v$.

**Proof strategy.** Given $D \in \Phi^{-1}(\mathbf{1})^+$, we define for all hidden neurons $v \in \mathcal{H}$:

$$\lambda_v := d_v > 0. \tag{39}$$

This definition automatically satisfies condition (i). To show that $D \in \mathcal{D}$, it therefore suffices to verify condition (ii), namely that for any edge $(u, v) \in E$:

$$d_{u \to v} = \frac{\lambda_v}{\lambda_u}. \tag{40}$$

We distinguish three cases based on the nature of the neurons $u$ and $v$. Note that we do not need to consider the case where $u \in N_{\text{in}}$ and $v \in N_{\text{out}}$ simultaneously, as such edges would bypass all hidden neurons and involve no rescaling from $\mathcal{D}$.

**Case 1:** $v \in N_{\text{out}}$

Consider the path $\mathrm{p} = (u \to v)$. By hypothesis, $\Phi(D) = \mathbf{1}$, so

$$\Phi_{\mathrm{p}}(D) = d_u \cdot d_{u \to v} = 1. \tag{41}$$

Since $d_u = \lambda_u$, we obtain

$$d_{u \to v} = \frac{1}{\lambda_u} \tag{42}$$

**Case 2:** $u \in N_{\text{in}}$

Consider a path $\mathrm{p}$ starting from input neuron $u$ and passing through the edge $(u, v)$. We can write $\mathrm{p} = (u \to v \to w_1 \to \cdots \to w_k)$ where $w_k \in N_{\text{out}}$. Consider also the path $\mathrm{p}' = (v \to w_1 \to \cdots \to w_k)$ which coincides with $\mathrm{p}$ after vertex $v$. By hypothesis, $\Phi(D) = \mathbf{1}$, we have

$$\Phi_{\mathrm{p}}(D) = d_{u \to v} \prod_{\ell=1}^{k} d_{w_{\ell-1} \to w_\ell} = 1, \tag{43}$$

where $w_0 := v$, and

$$\Phi_{\mathrm{p}'}(D) = d_v \prod_{\ell=1}^{k} d_{w_{\ell-1} \to w_\ell} = 1. \tag{44}$$

Dividing the first equation by the second yields

$$d_{u \to v} = d_v = \lambda_v. \tag{45}$$

**Case 3:** $u, v \in \mathcal{H}$

Let $(u, v) \in E$ be an edge between two hidden neurons.

Consider any path p starting from neuron $u$ and passing through the edge $(u, v)$. We can write $p = (u \to v \to w_1 \to \cdots \to w_k)$ where $w_k \in N_{\text{out}}$. Consider also the path $p' = (v \to w_1 \to \cdots \to w_k)$ which coincides with p after vertex $v$. Since both $u$ and $v$ are hidden neurons, both paths start from their respective biases. By hypothesis, $\Phi(D) = \mathbf{1}$, we have

$$\Phi_{\text{p}}(D) = d_u \cdot d_{u \to v} \prod_{\ell=1}^{k} d_{w_{\ell-1} \to w_\ell} = 1, \tag{46}$$

where $w_0 := v$, and

$$\Phi_{\text{p}'}(D) = d_v \prod_{\ell=1}^{k} d_{w_{\ell-1} \to w_\ell} = 1. \tag{47}$$

Dividing the first equation by the second yields

$$d_u \cdot d_{u \to v} = d_v. \tag{48}$$

Since $d_v = \lambda_v$ and $d_u = \lambda_u$, we obtain

$$d_{u \to v} = \frac{\lambda_v}{\lambda_u}. \tag{49}$$

In all cases, equation (40) holds, which shows that $D = \prod_{v \in \mathcal{H}} D_{\lambda_v, v} \in \mathcal{D}$. Therefore, $\Phi^{-1}(\mathbf{1})^+ \subseteq \mathcal{D}$.

$\square$

## E. Bregman matrix divergences

We first formally introduce Bregman matrix divergences. Let $S^q$ be the space of $q \times q$ symmetric matrices on $\mathbb{R}$. Given a strictly convex and differentiable function $\zeta : S^q \to \mathbb{R}$, the Bregman divergence between matrices $X$ and $Y$ is defined as

$$d_\zeta(X||Y) = \zeta(X) - \zeta(Y) - \text{tr}\left(\nabla\zeta(Y)^\top(X - Y)\right). \tag{50}$$

The most well-known example is built from the squared Frobenius norm $\zeta(X) = \|X\|_F^2$, which yields $d_\zeta(X||Y) = \|X - Y\|_F^2$. The logdet divergence corresponds to $\zeta(X) = -\log\det(X)$ (James et al., 1961), and the von Neumann divergence to $\zeta(X) = \text{tr}(X \log X - X)$ (Von Neumann, 1927). Within the class of Bregman matrix divergences, *spectral divergences* are particularly important: they take the form $\zeta(X) = \varphi \circ \lambda(X)$, where $\lambda$ denotes the vector of eigenvalues of $X$ in decreasing order and $\varphi$ is a strictly convex real-valued function on vectors (Kulis et al., 2009). The aforementioned examples are all spectral divergences: $\varphi(x) = \sum_i |x_i|^2$ for the Frobenius norm, $\varphi(x) = -\sum_i \log(x_i)$ for the logdet divergence, and $\varphi(x) = \sum_i(x_i \log x_i - x_i)$ for the von Neumann divergence.

Spectral divergences with $\varphi(x) = \sum_i \psi(x_i)$ for some $\psi : \mathbb{R} \to \mathbb{R}$ admit the expression (Kulis et al., 2009, Lemma 1) s

$$d_\zeta(X||Y) = \sum_{ij} (u_i^\top v_j)^2 (\psi(\lambda_i(X)) - \psi(\lambda_j(Y)) - (\lambda_i(X) - \lambda_j(Y))\nabla\psi(\lambda_j(Y)), \tag{51}$$

where $u_i$ (resp. $v_j$) is an eigenvector associated to the $i$-th largest eigenvalue $\lambda_i(X)$ of $X$ (resp. $Y$). Hence, $d_\zeta(X||Y)$ can be written as $d_\zeta(X||Y) = \sum_{ij}(u_i^\top v_j)^2 \text{div}_\psi(\lambda_i(X)||\lambda_j(Y))$ where $\text{div}_\psi$ is the Bregman divergence associated to $\psi$.

**Definition E.1.** Consider $\zeta = \varphi \circ \lambda$ with $\varphi(x) = \sum_i \psi(x_i)$ that induces a spectral divergence as described above. We define the $\zeta^+$ divergence as the divergence *that only considers the nonzero eigenvalues of the matrices*. In other words,

$$d_{\zeta^+}(X||Y) := \sum_{\substack{i:\lambda_i(X)\neq 0 \\ j:\lambda_j(Y)\neq 0}} (u_i^\top v_j)^2 \text{div}_\psi(\lambda_i(X)||\lambda_j(Y)). \tag{52}$$

We have the following lemma.

**Lemma E.2.** *Consider a spectral divergence on $S^q$ with $\zeta(X) = \varphi \circ \lambda(X)$ where $\varphi$ can be written as $\varphi(x) = \sum_i \psi(x_i)$ for some strictly convex and differentiable function $\psi : \mathbb{R} \to \mathbb{R}$ and $\zeta^+$ as defined in (52). Then, for any $A \in \mathbb{R}^{q \times p}$*

$$d_{\zeta^+}(AA^\top||I_q) = d_{\zeta^+}(A^\top A||I_p). \tag{53}$$

*Proof.* When $Y = I_q$, $d_{\zeta+}(X||Y)$ becomes $\sum_{i:\lambda_i(X)\neq 0} \|u_i\|_2^2 \mathrm{div}_\psi(\lambda_i(X)||1) = \sum_{i:\lambda_i(X)\neq 0} \mathrm{div}_\psi(\lambda_i(X)||1)$ thus only depends on the eigenvalues of $X$. Using that the nonzero eigenvalues of $AA^\top$ and $A^\top A$ are the same concludes. $\square$

Now we apply this result to the logdet divergence and we have

**Lemma E.3.** *Take $\zeta(X) = -\log\det(X)$. Then, for any $A \in \mathbb{R}^{q\times p}$,*

$$d_{\zeta+}(AA^\top||I_q) = \mathrm{tr}(A^\top A) - \sum_{i:\lambda_i(A^\top A)\neq 0} \log\lambda_i(A^\top A) - \mathrm{rank}(A)$$
$$= \mathrm{tr}(A^\top A) - \log\det_+(A^\top A) - \mathrm{rank}(A)\,,$$
(54)

*where $\det_+(X)$ it the generalized determinant, that is the product of the nonzeros eigenvalues of $X$.*

*Proof.* Based on the proof of Equation (52) we have $d_{\zeta+}(AA^\top||I_q) = \sum_{i:\lambda_i(A^\top A)\neq 0} \mathrm{div}_\psi(\lambda_i(A^\top A)||1)$ where $\psi = -\log$. So in this case $\mathrm{div}_\psi(x||y) = -\log(x)+\log(y)+\frac{1}{y}(x-y) = -\log(\frac{x}{y})+\frac{x}{y}-1$. Using this formula, $\mathrm{rank}(A^\top A) = \mathrm{rank}(A)$ and $\sum_{i:\lambda_i(A^\top A)\neq 0} \lambda_i(A^\top A) = \mathrm{tr}(A^\top A)$ gives the result. $\square$

## F. `PathCond` optimization problem

We detail here the definition of the `PathCond` optimization problem. We recall that $P_\theta = \partial\Phi(\theta)\partial\Phi(\theta)^\top \in \mathbb{R}^{q\times q}$ is the path-kernel. We define $G = G_\theta = \partial\Phi(\theta)^\top\partial\Phi(\theta) \in \mathbb{R}^{p\times p}$ and consider $\zeta(X) = -\mathrm{logdet}(X)$ and $d_\zeta, d_{\zeta+}$ the associated spectral divergence and its version that considers only nonzero eigenvalues, see Definition E.1. Furthermore, we place ourselves in the regime $q \geq p$ (which is the case for most DAG ReLU networks, except very small ones).

**Proposition F.1.** *Denote $r = \mathrm{rank}(\partial\Phi(\theta))$ and let $\partial\Phi(\theta) = U\Sigma V^\top$ be a compact SVD decomposition of $\partial\Phi(\theta)$, i.e., $U \in \mathbb{R}^{q\times r}, V \in \mathbb{R}^{p\times r}, U^\top U = V^\top V = I_r, \Sigma = \mathrm{diag}(\sigma_1,\cdots,\sigma_r) > 0$. For any admissible $D = \mathrm{diag}(d_1,\cdots,d_p) \in \mathcal{D}$ and $\alpha > 0$, we have*

$$d_{\zeta+}(\alpha P_{D\theta}||I_q) = d_{\zeta+}(\alpha G_{D\theta}||I_p) = \alpha\sum_{i=1}^p d_i^{-2}G_{ii} - r\log(\alpha) - \log\det(\Sigma^2) - \log\det(V^\top D^{-2}V) - r\,. \quad (55)$$

*When $r = p$, the matrix $G_{D\theta}$ is positive definite for any $D \in \mathcal{D}$ and*

$$d_{\zeta+}(\alpha P_{D\theta}||I_q) = d_{\zeta+}(\alpha G_{D\theta}||I_p) = d_\zeta(\alpha G_{D\theta}||I_p)\,. \quad (56)$$

*Moreover, the minimization over $\alpha > 0, D \in \mathcal{D}$ of these quantities is equivalent to the problem*

$$\min_{D'\in\mathcal{D}} p\log(\sum_{i=1}^p d_i'G_{ii}) - \sum_{i=1}^p \log(d_i')\,. \quad (57)$$

*Precisely if $D'$ solves (57) then $D := D'^{-1/2}$ and $\alpha = \frac{p}{\sum_i d_i^{-2}G_{ii}}$ minimize the previous quantities.*

*Proof.* First we recall that $\det_+$ is the product of the nonzero eigenvalues. First, we have from Lemma A.1 $\partial\Phi(D\theta) = \partial\Phi(\theta)D^{-1}$. Using this property, Lemma E.3 with $A = \sqrt{\alpha}\partial\Phi(\theta)D^{-1}$, and the commutation property of the trace, we have

$$d_{\zeta+}(\alpha P_{D\theta}||I_q) = d_{\zeta+}(\alpha\partial\Phi(\theta)D^{-1}(\partial\Phi(\theta)D^{-1})^\top||I_q)$$
$$= d_{\zeta+}(AA^\top||I_q)$$
$$= \mathrm{tr}(A^\top A) - \log\det_+(A^\top A) - \mathrm{rank}(A)$$
$$= \alpha\mathrm{tr}(D^{-2}G) - r\log(\alpha) - \log\det_+(D^{-1}GD^{-1}) - r\,.$$
(58)

Using the compact singular value decomposition of $\partial\Phi(\theta)$ we have that $D^{-1}GD^{-1} = D^{-1}V\Sigma^2V^\top D^{-1}$. Since $V$ has orthogonal columns and $D$ is invertible, the nonzero eigenvalues of $D^{-1}V\Sigma^2V^\top D^{-1}$ are the same as the nonzero eigenvalues of $\Sigma^2V^\top D^{-2}V$. Indeed, if $\lambda$ is such an eigenvalue then $D^{-1}V\Sigma^2V^\top D^{-1}x = \lambda x \implies V\Sigma^2V^\top D^{-1}x = \lambda Dx$. Thus $Dx \in \mathrm{Im}(V)$ and there is $y$ such that $Dx = Vy$ that is $x = D^{-1}Vy$. Thus, $V\Sigma^2V^\top D^{-1}D^{-1}Vy =$

$\lambda DD^{-1}Vy$ that is $V\Sigma^2 V^\top D^{-2}Vy = \lambda Vy$, left multiplying by $V^\top$ gives that $\lambda$ is an eigenvalue of $\Sigma^2 V^\top D^{-2}V$. The converse is straightforward.

Using the definition of $\det_+$ we get that

$$\det_+(D^{-1}GD^{-1}) = \det_+(\Sigma^2 V^\top D^{-2}V) = \det(\Sigma^2 V^\top D^{-2}V) = \det(\Sigma^2)\det(V^\top D^{-2}V)\,, \tag{59}$$

where we used that $\Sigma^2(V^\top D^{-2}V)$ is invertible since both $\Sigma$ and $V^\top D^{-2}V$ are invertible (the latter is positive definite because $D > 0$). This gives the first formula (55). Consequently, for any diagonal matrix $D$ we have

$$\min_{\alpha>0} d_{\zeta+}(\alpha P_{D\theta}||I_q) = -r\log(r) + r\log(\sum_{i=1}^p d_i^{-2}G_{ii}) - \log\det(\Sigma^2) - \log\det(V^\top D^{-2}V)\,, \tag{60}$$

as the first order conditions for the loss in $\alpha$ give $\alpha = \frac{r}{\sum_i d_i^{-2}G_{ii}}$ (the $G_{ii} \geq 0$ since the matrix is positive semi-definite).

When $r = p$, $G$ is full rank and since $G_{D\theta} = D^{-1}GD^{-1}$ it follows that $G_{D\theta}$ is positive definite hence $d_{\zeta+}(\alpha G_{D\theta}||I_p) = d_\zeta(\alpha G_{D\theta}||I_p)$ for any $\alpha > 0$ and any diagonal matrix $D$. In this case, since $V$ is square and unitary, $\log\det(V^\top D^{-2}V) = \log\det(D^{-2}) + \log\det(VV^\top) = \log\det(D^{-2})$. In light of (60), the minimization over $\alpha > 0$ and $D \in \mathcal{D}$ of $d_{\zeta+}(\alpha P_{D\theta}||I_q)$ thus corresponds to choosing $\alpha = \frac{r}{\sum_i d_i^{-2}G_{ii}}$ where $D \in \mathcal{D}$ minimizes the sum of terms that depend on $D$ in (60): $r\log\sum_{i=1}^p d_i^{-2}G_{ii} - \log\det(D^{-2})$. This corresponds to (57) with the change of variable $D' = D^{-2}$ (observe that $D' \in \mathcal{D}$ if, and only if, $D \in \mathcal{D}$) and the expression of the determinant of a diagonal matrix. $\qquad\square$

*Remark* F.2. With the same reasoning, when $r = \mathrm{rank}(\partial\Phi(\theta)) < p$ the minimization of (55) over $\alpha, D$ is equivalent to

$$\min_{D'\in\mathcal{D}} r\log(\sum_{i=1}^p d_i'G_{ii}) - \log\det(V^\top D'V)\,. \tag{61}$$

We now prove that the optimization problem (57) can be rewritten as (9) and that it has a solution.

**Lemma F.3.** *Consider $B \in \mathbb{R}^{p\times H}$ with entries in $\{1, -1, 0\}$ indicating whether a parameter enters $(-1)$, leaves $(1)$, or is unrelated to $(0)$ a given neuron. Precisely, $B = (b_1, \cdots, b_H)$ with*

$$\forall \text{ neuron } h,\ (b_h)_i = \begin{cases} -1 & \text{if } i \in \mathrm{in}_h \\ 1 & \text{if } i \in \mathrm{out}_h \\ 0 & \text{otherwise.} \end{cases} \tag{62}$$

*where $\mathrm{in}_h$ and $\mathrm{out}_h$ are formally defined in Definition D.1.*

*Then $\mathcal{D} = \{\mathrm{diag}(\exp(Bu)), u \in \mathbb{R}^H\}$. Thus (57) is equivalent to*

$$\min_{u\in\mathbb{R}^H} F(u) \quad \text{with } F(u) := p\log(\sum_{i=1}^p e^{(Bu)_i}G_{ii}) - \sum_{i=1}^p (Bu)_i\,, \tag{63}$$

*in the sense that if $u$ solves (63) then $D' := \mathrm{diag}(\exp(Bu))$ solves (57).*

*Proof.* Let $D' \in \mathcal{D}$, and $d$ be the vector of $\mathbb{R}^p$ such that $D' = \mathrm{diag}(d)$. We want to show that $d$ can be written as $d = \exp(Bu)$. Thanks to Lemma D.3, we know that there exists $\boldsymbol{\lambda} \in (\mathbb{R}^{+*})^H$ such that

$$d = \bigodot_{v\in\mathcal{H}} d_{\lambda_v,v} \tag{64}$$

Where $\bigodot$ denotes the Hadamard coordinate wise product and $d_{\lambda_v,v}$ is the diagonal of the neuron-wise rescaling matrix $D_{\lambda_v,v}$ introduced in Definition D.1. For $v \in \mathcal{H}$, $\lambda_v \in \mathbb{R}^{+*}$, by definition, we get

$$(d_{\lambda_v,v})_i = \begin{cases} \lambda_v & \text{if } i \in \mathrm{in}_v \\ 1/\lambda_v & \text{if } i \in \mathrm{out}_v \\ 1 & \text{otherwise.} \end{cases} \tag{65}$$

But because $\lambda_v \in \mathbb{R}^{+*}$, there exists an $u_v \in \mathbb{R}$ such that $\lambda_v = \exp(-u_v)$. We can rewrite $d_{\lambda_v,v}$ as

$$(d_{\lambda_v,v})_i = \begin{cases} \exp(-u_v) & \text{if } i \in \text{in}_v \\ \exp(u_v) & \text{if } i \in \text{out}_v \\ \exp(0) & \text{otherwise.} \end{cases} \tag{66}$$

We can rewrite the previous equation with a column of $B$, precisely:

$$d_{\lambda_v,v} = \exp(u_v b_v) \tag{67}$$

where the $\exp$ is taken coordinate-wise. Now let $i$ an integer with $1 \le i \le p$, thanks to (64), we get

$$d_i = \prod_{h=1}^{H} (d_{\lambda_h,h})_i = \prod_{h=1}^{H} \exp(u_h b_h)_i = \prod_{h=1}^{H} \exp(B_{ih} u_h) \tag{68}$$

So,

$$d_i = \exp\Big(\sum_{h=1}^{H} B_{ih} u_h\Big) \tag{69}$$

We can conclude that

$$d = \exp(Bu) \tag{70}$$

Vice-versa, if $d$ writes this way it is a Hadamard product of $d_h := \exp(b_h u_h)$, so that $D' = \text{diag}(d)$ is a product of diagonal matrices which can easily be checked to belong to $\mathcal{D}$. We conclude using that $\mathcal{D}$ is a group. $\qquad\square$

To prove existence of a solution we will use the following result.

**Lemma F.4.** *If $G_{ii} > 0$ for each $i$ and $B \ne 0$ satisfies $1 \notin \text{span}(B)$, then the function $F$ in (63) is coercive.*

*Proof.* $F$ can be written as $F(u) = f(Bu)$ where $f(x) = p \log(\sum_{i=1}^{p} e^{x_i} G_{ii}) - \sum_i x_i$. We have that $\sum_{i=1}^{p} e^{x_i} G_{ii} \ge \max_{k \in [\![p]\!]} e^{x_k} G_{kk}$ thus $f(x) \ge p \max_{k \in [\![p]\!]}\{x_k + \log(G_{kk})\} - \sum_i x_i \ge pc + \sum_i(\max_k x_k - x_i)$ where $c = \min_k \log(G_{kk}) > -\infty$ using the hypothesis on $G$. Denoting $e_i$ the i-th standard basis vector, this gives

$$F(u) \ge pc + \sum_{i=1}^{p} \Big( \max_k\{\langle e_k^\top, Bu \rangle\} - \langle e_i^\top, Bu \rangle \Big). \tag{71}$$

We define

$$\forall u, \ \alpha(u) = \sum_{i=1}^{p} \Big( \max_k\{\langle e_k^\top, Bu \rangle\} - \langle e_i^\top, Bu \rangle \Big). \tag{72}$$

Clearly for any $t \ge 0$, $\alpha(t \cdot u) = t \cdot \alpha(u)$. To conclude that the function is coercive we introduce

$$\beta := \min_{\|u\|_2 = 1} \alpha(u). \tag{73}$$

Since $\alpha$ is continuous, this minimum is attained. Also, for any $u \ne 0$,

$$F(u) = F\left( \|u\|_2 \cdot \frac{u}{\|u\|_2} \right) \ge pc + \alpha\left( \|u\|_2 \cdot \frac{u}{\|u\|_2} \right) = pc + \|u\|_2 \cdot \alpha\left( \frac{u}{\|u\|_2} \right) \ge pc + \beta\|u\|_2. \tag{74}$$

Proving that $\beta > 0$ is thus sufficient to ensure the coercivity of $F$. For this, first note that for any $u, i \in [\![p]\!]$,

$$\max_k\{\langle e_k^\top, Bu \rangle\} - \langle e_i^\top, Bu \rangle \ge 0, \tag{75}$$

hence $\beta \ge 0$, and we only need to rule out the case $\beta = 0$. The above inequality indeed also shows $\beta = 0$ if and only if there exists $u$ such that $\forall i, \max_k\{\langle e_k^\top, Bu \rangle\} = \langle e_i^\top, Bu \rangle$. This condition is equivalent to the existence of $u$ such that $\langle e_i^\top, Bu \rangle = \text{cte}, \forall i$, that is to say the existence of $u$ such that $Bu = \text{cte}$, which is not possible since $1 \notin \text{span}(B)$. $\qquad\square$

To conclude we use the second lemma

**Lemma F.5.** *Consider $B$ as defined in* (62). *Then* $1 \notin \mathrm{span}(B)$.

*Proof.* By contradiction, assume that $1 \in \mathrm{span}(B)$ then there is $u$ such that $Bu = 1$, and by Lemma F.3 the diagonal matrix $D := \mathrm{diag}(\exp(Bu)) = e^1 I$ belongs to $\mathcal{D}$ and thanks to Lemma D.7 it belongs to $\Phi^{-1}(\mathbf{1})^+$. However for any path $\mathrm{p} \in \mathrm{P}$, $\Phi_{\mathrm{p}}(e^1 I) = \prod_{i \in p} e^1 \neq 1$ which leads to the desired contradiction. We conclude that $1 \notin \mathrm{span}(B)$. $\qquad\square$

**Corollary F.6.** *Suppose that* $\forall i,\ G_{ii} > 0$. *The optimization problem* (9) *always has a solution.*

*Proof.* Since the logsumexp function is convex (Gao & Pavel, 2017, Lemma 4), one easily shows that the function $F$ is convex. It is also continuous, and coercive by Lemma F.4 and Lemma F.5. So there exists a minimizer. $\qquad\square$

Finally we prove that (9) can be solved via simple alternating minimization, where each minimization admits a closed form.

**Lemma F.7.** *We note* $\rho_{i,h} = e^{(Bu)_i - B_{ih} u_h}$. *If* $g > 0$, *for any neuron* $h$, *the problem* $\min_{u_h \in \mathbb{R}} F(u_1, \cdots, u_h, \cdots, u_H)$ *has a solution given by* $\log(r_h)$ *where* $r_h$ *is the unique positive root of the polynomial* $B(A+p)X^2 + ADX + C(A-p)$ *with*

$$\mathcal{A} := |\{i \in \mathrm{in}_h\}| - |\{i \in \mathrm{out}_h\}|, \quad \mathcal{B} := \sum_{i \in \mathrm{out}_h} \rho_{i,h} g_i$$

$$\mathcal{C} := \sum_{i \in \mathrm{in}_h} \rho_{i,h} g_i, \quad \mathcal{D} := \sum_{i \in \mathrm{other}_h} \rho_{i,h} g_i \,.$$

*Proof.* We note, $g_i = G_{ii}$, $f(x) = p \log(\sum_i e^{x_i} g_i) - \sum_i x_i$ such that $F(u) = f(Bu)$. We have first

$$\nabla f(x) = \frac{p}{\langle e^x \odot g, 1 \rangle} (e^x \odot g) - 1 \,. \tag{76}$$

Since $\nabla F(u) = B^\top \nabla f(Bu)$ the $h$-th coordinate of this gradient is

$$
\begin{aligned}
(\nabla F(u))_h = \langle b_h, \nabla f(Bu) \rangle &= \frac{p}{\langle e^{Bu} \odot g, 1 \rangle} \langle b_h, e^{Bu} \odot g \rangle - \langle b_h, 1 \rangle \\
&= p \frac{\sum_{i \in \mathrm{out}_h} e^{(Bu)_i} g_i - \sum_{i \in \mathrm{in}_h} e^{(Bu)_i} g_i}{\sum_i e^{(Bu)_i} g_i} + \Big( \sum_{i \in \mathrm{in}_h} 1 - \sum_{i \in \mathrm{out}_h} 1 \Big) \,.
\end{aligned} \tag{77}
$$

Now writing $u = (u_1, \cdots, u_h, \cdots, u_H)$ and using that

$$(Bu)_i = \sum_k B_{ik} u_k = B_{ih} u_h + \sum_{k \neq h} B_{ik} u_k \,, \tag{78}$$

and with the notation $\rho_{i,h} := \exp((Bu)_i - B_{ih} u_h)$, we get

$$
e^{(Bu)_i} = \rho_{i,h} e^{B_{ih} u_h} = \begin{cases} \rho_{i,h} e^{u_h} & i \in \mathrm{out}_h \\ \rho_{i,h} e^{-u_h} & i \in \mathrm{in}_h \\ \rho_{i,h} & i \in \mathrm{other}_h \,. \end{cases} \tag{79}
$$

So with $A, B, C, D$ defined in the lemma, (77) becomes

$$
\begin{aligned}
(\nabla F(u))_h &= p \frac{\sum_{i \in \mathrm{out}_h} e^{u_h} \rho_{i,h} g_i - \sum_{i \in \mathrm{in}_h} e^{-u_h} \rho_{i,h} g_i}{\sum_{i \in \mathrm{out}_h} e^{u_h} \rho_{i,h} g_i + \sum_{i \in \mathrm{in}_h} e^{-u_h} \rho_{i,h} g_i + \sum_{i \in \mathrm{other}_h} \rho_{i,h} g_i} + A \\
&= p \frac{e^{u_h} \mathcal{B} - e^{-u_h} \mathcal{C}}{e^{u_h} \mathcal{B} + e^{-u_h} \mathcal{C} + \mathcal{D}} + \mathcal{A} \,.
\end{aligned} \tag{80}
$$

Hence

$$(\nabla F(u))_h = 0 \iff \mathcal{B}(\mathcal{A}+p)e^{u_h} + \mathcal{A}\mathcal{D} + \mathcal{C}(\mathcal{A}-p)e^{-u_h} = 0 \,. \tag{81}$$

Introducing $X = e^{u_h} > 0$, the problem of finding a minimizer $u_h$ (which we know exists by coercivity of $F$) reduces to finding a positive root of the polynomial

$$\mathcal{B}(\mathcal{A} + p)X^2 + \mathcal{A}\mathcal{D}X + \mathcal{C}(\mathcal{A} - p). \tag{82}$$

One easily checks that $-p < \mathcal{A} < p$ and $\mathcal{B}, \mathcal{C} > 0$ (since $g > 0$), hence $\mathcal{B}(\mathcal{A} + p) > 0, \mathcal{C}(\mathcal{A} - p) < 0$ thus the polynomial has exactly one positive root. $\qquad\square$

## G. Computation of the diagonal of $G$

**Proposition G.1.** *The diagonal of $G = \partial\Phi^\top \partial\Phi$ can be computed efficiently as:*

$$\mathrm{diag}(G) = \nabla_{\theta^2}\|\Phi(\theta)\|_2^2 \tag{83}$$

*where $\nabla_{\theta^2}$ denotes differentiation with respect to the vector of squared parameters $\theta^2 = (\theta_1^2, \ldots, \theta_p^2)$, or equivalently $\theta^2 = \theta \odot \theta$ where $\odot$ denotes the Hadamard (element-wise) product and $p$ is the number of parameters.*

*Proof.* Denote P the set of all paths. For a parameter $i$, we have by making the change of variable $\theta_i^2 = \omega_i$ :

$$G_{ii} = \sum_{\substack{p\in P \\ p\ni i}} \prod_{\substack{k\in p \\ k\neq i}} \theta_k^2 \qquad \text{(definition of } G_{ii}) \tag{84}$$

$$= \sum_{\substack{p\in P \\ p\ni i}} \prod_{\substack{k\in p \\ k\neq i}} \omega_k \tag{85}$$

$$= \sum_{\substack{p\in P \\ p\ni i}} \frac{\partial}{\partial\omega_i}\Phi_p(\omega) \qquad \text{(Because } \Phi_p(\omega) = \prod_{k\in p}\omega_k\text{, the derivative is straightforward)} \tag{86}$$

$$= \frac{\partial}{\partial\omega_i} \sum_{p\in P}\Phi_p(\omega) \qquad \text{(terms with } i \notin p \text{ do not depend on } \omega_i \text{ and vanish with the derivative)} \tag{87}$$

$$= (\nabla_\omega\|\Phi(\omega)\|_1)_i \qquad \text{(Because } \Phi_p(\omega) \geq 0) \tag{88}$$

$$= (\nabla_{\theta^2}\|\Phi(\theta)\|_2^2)_i \qquad \text{(Because } \|\Phi_p(\omega)\|_1 = \|\Phi_p(\theta)\|_2^2) \tag{89}$$

$\qquad\square$

## H. Detailed Algorithm and Complexity Analysis

Algorithm 3 gives the full pseudocode of `PathCond`. The matrix $B \in \mathbb{R}^{p\times H}$ (defined in 62) is never formed explicitly; only its non-zero entries are stored. Since each parameter (edge) appears in at most two columns of $B$ (once as an outgoing edge and once as an incoming edge) we have $\mathrm{nnz}(B) \leq 2p$. The $O(n_{\mathrm{iter}} \cdot p)$ total complexity rests on two incremental updates.

**Dual variable $v = Bu$.** At each coordinate-descent step, only $u_h$ changes by some increment $\delta$, inducing the update $\Delta v = \delta\, b_h$, where $b_h$ is the $h$-th column of $B$. Since $b_h$ has only $|\mathrm{out}_h| + |\mathrm{in}_h|$ non-zero entries, this costs $O(|\mathrm{out}_h| + |\mathrm{in}_h|)$ rather than $O(p)$.

**Global auxiliary sum $E$.** Computing the degree polynomial (Lemma 4.1) requires

$$\mathcal{D} = \sum_{i\in\mathrm{other}_h} e^{v_i}\, g_i,$$

a sum over $p - |\mathrm{out}_h| - |\mathrm{in}_h| \approx p$ terms. Naïvely evaluating this for each of the $H$ neurons would give $O(pH)$ per sweep. Instead, we maintain the global sum $E = \sum_{i=1}^p e^{v_i} g_i$ and recover

$$\mathcal{D} = E - S_{\mathrm{out}} - S_{\mathrm{in}}, \qquad S_{\mathrm{out}} = \sum_{i\in\mathrm{out}_h} e^{v_i}g_i, \qquad S_{\mathrm{in}} = \sum_{i\in\mathrm{in}_h} e^{v_i}g_i,$$

---

**Algorithm 3** `PathCond` : Path-conditioned rescaling

---

**Input** : DAG ReLU network with parameters $\theta$; scaling weights $g = \text{diag}(G) \in \mathbb{R}^p$; tolerance $\varepsilon > 0$; maximum iterations $n_{\text{iter}}$

**Output** : Rescaled parameters $\tilde{\theta} = \text{diag}(e^v)\,\theta$

---

7  $u \leftarrow \mathbf{0} \in \mathbb{R}^H$;                                                    // Primal variables

8  $v \leftarrow \mathbf{0} \in \mathbb{R}^p$;                                        // Dual variables, invariant: $v = Bu$

9  $E \leftarrow \sum_{i=1}^{p} g_i$;                                // Global sum $E = \sum_i e^{v_i} g_i$, initialized with $v = 0$

10  **for** $k = 0, \dots, n_{\text{iter}} - 1$ **do**

11  $\quad$ $\Delta \leftarrow 0$ **for** *each neuron* $h = 1, \dots, H$ **do**

$\quad\quad$ // Index sets of outgoing/incoming parameters of neuron $h$

12  $\quad\quad$ $\text{out}_h \leftarrow \{i : B_{i,h} = +1\}$ $\text{in}_h \leftarrow \{i : B_{i,h} = -1\}$

$\quad\quad$ // Recover $\mathcal{D}$ from global sum in $O(|\text{out}_h| + |\text{in}_h|)$

13  $\quad\quad$ $S_{\text{out}} \leftarrow \sum_{i \in \text{out}_h} e^{v_i} g_i$ $S_{\text{in}} \leftarrow \sum_{i \in \text{in}_h} e^{v_i} g_i$ $\mathcal{D} \leftarrow E - S_{\text{out}} - S_{\text{in}}$

$\quad\quad$ // Coefficients of the degree polynomial (Lemma 4.1)

14  $\quad\quad$ $\mathcal{A} \leftarrow |\text{in}_h| - |\text{out}_h|$ $\mathcal{B} \leftarrow \sum_{i \in \text{out}_h} e^{v_i - u_h}\, g_i$;              // $\rho_{i,h} = e^{v_i - u_h}$ for $i \in \text{out}_h$

15  $\quad\quad$ $\mathcal{C} \leftarrow \sum_{i \in \text{in}_h} e^{v_i + u_h}\, g_i$;              // $\rho_{i,h} = e^{v_i + u_h}$ for $i \in \text{in}_h$

$\quad\quad$ // Optimal step: solve $\mathcal{B}(\mathcal{A}+p)\,X^2 + \mathcal{A}\mathcal{D}\,X + \mathcal{C}(\mathcal{A}-p) = 0$

16  $\quad\quad$ $r_h \leftarrow \dfrac{-\mathcal{A}\mathcal{D} + \sqrt{(\mathcal{A}\mathcal{D})^2 - 4\,\mathcal{B}(\mathcal{A}+p)\,\mathcal{C}(\mathcal{A}-p)}}{2\,\mathcal{B}(\mathcal{A}+p)}$ $\delta \leftarrow \ln r_h - u_h$;              // $u_h^{(k+1)} = \ln r_h$

$\quad\quad$ // Incremental update of $v$ and $E$ in $O(|\text{out}_h| + |\text{in}_h|)$

17  $\quad\quad$ $v_i \leftarrow v_i + \delta$ for $i \in \text{out}_h$ $v_i \leftarrow v_i - \delta$ for $i \in \text{in}_h$ $E \leftarrow \mathcal{D} + \sum_{i \in \text{out}_h} e^{v_i} g_i + \sum_{i \in \text{in}_h} e^{v_i} g_i$

18  $\quad\quad$ $u_h \leftarrow u_h + \delta$ $\Delta \leftarrow \Delta + |\delta|$

19  $\quad$ **if** $\Delta < \varepsilon$ **then break**;

20  **return** $\tilde{\theta} = \text{diag}(e^v)\,\theta$

---

at cost $O(|\text{out}_h| + |\text{in}_h|)$. After updating $v$, the sum $E$ is refreshed within the same budget by recomputing only the affected terms. One full sweep therefore costs $\sum_{h=1}^{H} O(|\text{out}_h| + |\text{in}_h|) = O(p)$, and the total complexity over $n_{\text{iter}}$ sweeps is $O(n_{\text{iter}} \cdot p)$, linear in the number of parameters per iteration.

### H.1. Computational Cost: Additional Results

This section provides additional timing measurements and computational analysis complementing Section 5.3 of the main paper. The `PathCond` algorithm consists of four main computational steps:

1. **Diagonal of $G$:** computing $\text{diag}(G)$ where $G = \partial\Phi^\top \partial\Phi$ (Section G), which requires a single backward pass.

2. **Connectivity matrix $B$:** computing $B \in \mathbb{R}^{p \times H}$ with entries in $\{-1, 0, 1\}$, indicating whether a parameter enters $(-1)$, leaves $(+1)$, or is unrelated $(0)$ to a given hidden neuron (Definition D.1).

3. **Block coordinate descent (BCD):** solving the optimization problem with $n_{\text{iter}}$ passes of BCD to obtain the optimal rescaling matrix.

4. **Rescaling:** applying the computed rescaling to update model parameters in place.

Figure 7 (left) reports the rescaling time of `PathCond` against the cost of a single CIFAR-10 training epoch, for MLPs of varying depth ($L \in \{2, 3, \dots, 9\}$) and constant hidden width (500). It demonstrates that rescaling time scales linearly with the number of parameters, as expected from the $O(\#\text{params})$ complexity of computing the diagonal and the BCD iterations. Figure 7 (right) decomposes the wall-clock time by step for three architectures. Steps 1, 2, and 4 are negligible compared to step 3 (BCD), which accounts for roughly $98\%$ of the total rescaling time. The BCD step is therefore the primary target for future optimizations.

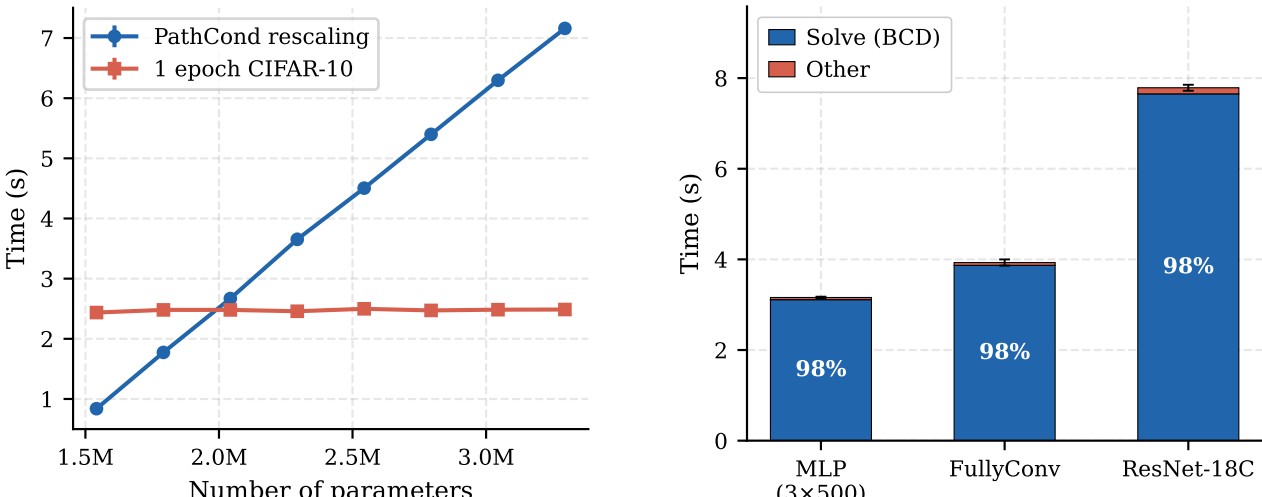

*Figure 7.* Timing of `PathCond` rescaling. (**Left**) Total rescaling time for MLPs with varying depth ($L \in \{2, 3, \ldots, 9\}$) and constant hidden width (500). (**Right**) `PathCond` time decomposition for three architectures. Error bars denote standard deviations over 10 runs.

## H.2. Number of BCD Iterations: Convergence Analysis

A natural question is how many BCD iterations are needed for a "good" rescaling. While `PathCond` has no hyperparameters in the usual sense, the number of iterations $n_{\text{iter}}$ controls a trade-off between computational cost and solution quality. We present an ablation showing that $n_{\text{iter}} = 10$ (the value used in all our experiments) is a good compromise, even though the convergence rate varies across architectures.

**Experimental setup.** For each architecture (an MLP with BatchNorm (depth $L = 4$, hidden width 500), the CIFAR-NV fully convolutional network, and ResNet-18C) and for $n_{\text{iter}} \in \{1, \ldots, 100\}$, we monitor:

1. **Normalized objective:** $\left(F(u^{(k)}) - F(u^\star)\right)/\left(F(u^{(0)}) - F(u^\star)\right)$, where $F$ is the objective of Equation (9), $u^{(0)} = 0$ is the initialization, and $u^\star \approx u^{(100)}$ approximates the optimum. The normalized objective lies in $[0, 1]$ and equals 0 at convergence.

2. **Mean primal update:** $\frac{1}{H}\|u^{(k+1)} - u^{(k)}\|_1$, the average per-neuron change in the primal variable $u$ between consecutive iterations. Recall that the rescaling factors are recovered as $\lambda_h = \pm e^{u_h}$; small values therefore signal BCD convergence.

**Results.** Figure 8 (left) shows that convergence speed depends strongly on the architecture. For the simple MLP, the objective reaches about 90% of its final decrease within 5 iterations. The fully convolutional network and ResNet-18C converge more slowly: at 10 iterations they have only realized 20–30% of the total objective decrease, and 25–30 iterations are needed to approach convergence. Figure 8 (right) corroborates this picture from the primal iterates: after 20 iterations, the average per-neuron update falls below $10^{-2}$ for the MLP, but remains of order $10^{-1}$ for FullyConv and ResNet-18C, reflecting the different conditioning and structure of these architectures.

In light of this analysis, we set $n_{\text{iter}} = 10$ as a practical compromise: it yields near-optimal solutions for simpler architectures and substantial improvements for more complex ones, while keeping the computational overhead small (typically half to one CIFAR-10 epoch) across all architectures.

## I. Architecture Statistics: Number of Parameters, Hidden Units, and Paths

This section reports detailed architectural statistics for the neural networks used in our experiments. All networks take a CIFAR-10 image as input, i.e. a tensor of shape $3 \times 32 \times 32$. For each architecture, we report:

- **Parameters:** the total number of trainable parameters (weights and biases).

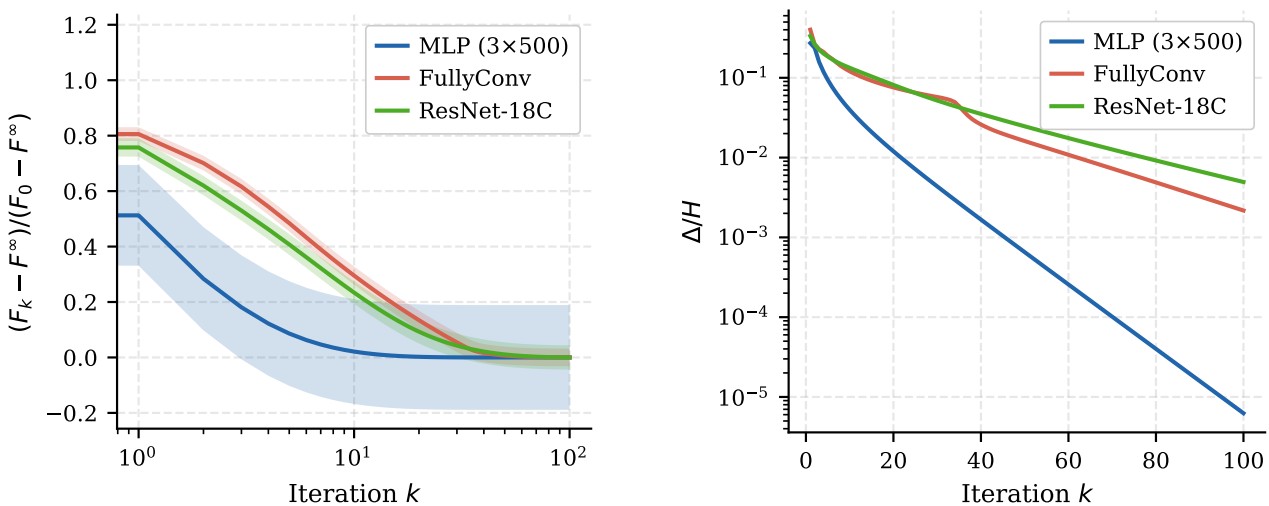

*Figure 8.* BCD convergence. (**Left**) Normalized objective $F(u^{(k)})$ as a function of the iteration index. (**Right**) Mean primal step size $\|u^{(k+1)} - u^{(k)}\|_1/H$ across iterations. Curves show the mean and standard deviation over 10 seeds.

- **Hidden Units:** the total number of hidden channels across all convolutional layers (the convolutional analogue of hidden neurons in fully connected networks), or simply the total number of hidden neurons for MLPs. The final classification layer is excluded.

- **Number of Paths:** the total number of distinct computational paths from input pixels to output logits induced by the architecture.

To compute the number of paths, we rely on the path-norm formalism of Gonon et al. (2023). Let P denote the set of all paths and let $\Phi_\mathrm{p}(\theta)$ denote the product of parameters along path p. The path-1 norm is

$$\|\Phi(\theta)\|_1 \;=\; \sum_{\mathrm{p}\in\mathrm{P}} \big|\Phi_\mathrm{p}(\theta)\big|.$$

Setting all parameters to one ($\theta = \mathbf{1}$), each path product equals one, and therefore

$$\|\Phi(\mathbf{1})\|_1 \;=\; \sum_{\mathrm{p}\in\mathrm{P}} 1 \;=\; |\mathrm{P}|,$$

which is exactly the number of paths in the network. This quantity is computed efficiently via a single forward pass on a CIFAR-10-shaped input with parameters set to one (Gonon et al., 2023).

Table 1 summarizes these statistics for several architectures.

| Model | Parameters | Hidden Units | Paths |
|---|---|---|---|
| MLP $3 \times 500$ | 2,042,510 | 1,500 | $3.84 \times 10^{12}$ |
| FullyConv (CIFAR-NV) (Gitman & Ginsburg, 2017) | 2,616,576 | 1,792 | $4.13 \times 10^{26}$ |
| ResNet-18 (He et al., 2016) | 11,689,512 | 4,800 | $2.24 \times 10^{53}$ |
| ResNet-34 (He et al., 2016) | 21,797,672 | 8,512 | $1.19 \times 10^{100}$ |
| ResNet-50 (He et al., 2016) | 25,557,032 | 26,560 | $2.64 \times 10^{136}$ |
| VGG-16 (Simonyan & Zisserman, 2014) | 138,357,544 | 12,416 | $5.89 \times 10^{54}$ |

*Table 1.* Architectural statistics for the networks used in this work. All models take a CIFAR-10 image ($3 \times 32 \times 32$) as input.

## J. Regimes Analysis at Initialization

We begin with a fundamental observation about the rescaling algorithm.

**Proposition J.1** (Trivial case: constant diagonal). *If the diagonal of $G$ is constant, i.e., there exists $\alpha > 0$ such that*

$$g = \alpha \mathbf{1},$$

*then the objective of the rescaling problem* (9) *admits $u = 0$ as its unique minimizer. Consequently, the rescaling matrix is the identity and the algorithm has no effect.*

*Proof.* Let's go back to (9)

$$F(u) = p \log \left( \sum_{i=1}^{p} e^{(Bu)_i} G_{ii} \right) - \sum_{i=1}^{p} (Bu)_i.$$

If $G_{ii} = \alpha > 0$ for all $i$, then

$$F(u) = p \log \left( \frac{\alpha}{p} \sum_{i=1}^{p} e^{(Bu)_i} \right) - \sum_{i=1}^{p} (Bu)_i = p \log \alpha + p \log \left( \frac{1}{p} \sum_{i=1}^{p} e^{(Bu)_i} \right) - p \left( \frac{1}{p} \sum_{i=1}^{p} (Bu)_i \right).$$

Because the $\log$ is concave on $\mathbb{R}_*^+$, by Jensen's inequality, we have

$$F(u) \geq p \log(\alpha),$$

which is achieved at $u = 0$. Since the objective is strictly convex, $u = 0$ is the unique minimizer. □

### J.1. The Matrix $G$

Consider a neural network represented as a directed acyclic graph (DAG) with ReLU, identity, and max-pooling operations. Let $\Phi$ denote the path lifting map. We have the following matrix:

$$G := \partial \Phi^\top \partial \Phi.$$

We focus on its diagonal $g = \mathrm{diag}(G)$. For any parameter $i$ (edge weight or bias), the corresponding diagonal entry is

$$g_i = \sum_{\substack{p \in P \\ p \ni i}} \prod_{\substack{j \in p \\ j \neq i}} \theta_j^2, \tag{90}$$

where $P$ denotes the set of all paths ending at an output neuron.

**Expectation of $\mathrm{diag}(G)$** At initialization, parameters $\theta_i$ are independent random variables. By independence, the linearity of expectation yields

$$\mathbb{E}[g_i] = \sum_{\substack{p \in P \\ p \ni i}} \prod_{\substack{j \in p \\ j \neq i}} \mathbb{E}[\theta_j^2]. \tag{91}$$

### J.2. Layered Fully-Connected ReLU Networks

We specialize to layered fully-connected ReLU networks (LFCNs) (Gonon et al., 2023). Such a network is specified by a depth $L$ and widths $(n_0, \ldots, n_L)$, where layer $k$ maps $\mathbb{R}^{n_k}$ to $\mathbb{R}^{n_{k+1}}$.

We distinguish two types of paths: (i) paths starting from input neurons, and (ii) paths starting from bias parameters. As is customary, we omit a bias in the last layer since it does not affect expressivity.

Let $\sigma_k^2 := \mathbb{E}[\theta_j^2]$ for any edge parameter in layer $k$. This assumption of layer-wise constant variance is standard in neural network initialization schemes such as Xavier/Glorot initialization (Glorot & Bengio, 2010) and He initialization (He et al., 2015), which are designed to maintain activation variance across layers during forward and backward passes.

**Counting paths through a parameter.** To compute expected path magnitudes via equation (91), we decompose the calculation into: (1) counting paths passing through a parameter, and (2) computing the expected squared magnitude of each path type.

Consider an edge parameter $w$ located in layer $k$ (connecting layer $k$ to layer $k+1$). The number of paths passing through $w$ and originating from input neurons equals the number of input-to-output paths in the reduced network obtained by fixing layers $k$ and $k+1$ to a single neuron:

$$N_{\text{edge},k}^{\text{input}} = \prod_{\substack{j=0 \\ j \neq k,k+1}}^{L} n_j.$$

This counts all combinations of neurons in layers other than $k$ and $k+1$.

For any such path p containing $w$, the expected contribution to $g_w$ (excluding $w$ itself) is

$$\mathbb{E}\left[\prod_{\substack{j \in \text{p} \\ j \neq w}} \theta_j^2\right] = \prod_{\substack{m=0 \\ m \neq k}}^{L-1} \sigma_m^2,$$

where the product ranges over all layers except layer $k$ (which contains $w$) and the output layer $L$ (which has no outgoing edges).

Additionally, we must account for paths originating from bias parameters in layers $0$ to $k-1$. A path starting from a bias in layer $j < k$ and passing through $w$ traverses all layers from $j+1$ to $L$, excluding the neurons in layers $k$ and $k+1$. The number of such paths is

$$N_{\text{edge},k}^{\text{bias},j} = \prod_{\substack{m=j+1 \\ m \neq k,k+1}}^{L} n_m.$$

Each such path has expected squared magnitude (excluding $w$ from the product)

$$\prod_{\substack{m=i \\ m \neq k}}^{L-1} \sigma_m^2,$$

where the product starts from layer $i$ (containing the bias) and excludes layer $k$.

For a bias parameter $b$ in layer $k$, the analysis is simpler. Such a bias only participates in paths from layer $k$ onward to the output. The number of such paths is

$$N_{\text{bias},k} = \prod_{j=k+2}^{L} n_j,$$

and each path has expected squared magnitude (excluding the bias itself from the product)

$$\prod_{m=k+1}^{L-1} \sigma_m^2.$$

**Expected path magnitudes: closed-form formula.** Combining path counts with their corresponding expected squared magnitudes yields the following closed-form expression for expected path magnitudes:

$$\mathbb{E}[g_i] = \begin{cases} \displaystyle\prod_{j=k+2}^{L} n_j \prod_{m=k+1}^{L-1} \sigma_m^2, & \text{if } i \text{ is a bias in layer } k, \\[2em] \displaystyle\prod_{\substack{j=0 \\ j \neq k,k+1}}^{L} n_j \prod_{\substack{m=0 \\ m \neq k}}^{L-1} \sigma_m^2 + \sum_{j=0}^{k-1} \prod_{\substack{m=j+1 \\ m \neq k,k+1}}^{L} n_m \prod_{\substack{m=j \\ m \neq k}}^{L-1} \sigma_m^2, & \text{if } i \text{ is an edge in layer } k. \end{cases} \tag{92}$$

Introducing the quantities $a_k := n_k \sigma_k^2$, equation (92) simplifies to:

$$
\mathbb{E}[g_i] = 
\begin{cases}
n_L \sigma_{k+1}^2 \displaystyle\prod_{j=k+2}^{L-1} a_j, & \text{if } i \text{ is a bias in layer } k, \\[2em]
n_L \sigma_{k+1}^2 \displaystyle\prod_{\substack{j=0 \\ j \neq k, k+1}}^{L-1} a_j + n_L \sigma_{k+1}^2 \sum_{j=0}^{k-1} \sigma_j^2 \prod_{\substack{m=j+1 \\ m \neq k}}^{L-1} a_m, & \text{if } i \text{ is an edge in layer } k.
\end{cases}
\tag{93}
$$

### J.3. Classical Initializations

For standard initializations such as He or Kaiming initialization (He et al., 2015), one has

$$
a_k = n_k \sigma_k^2 = a, \quad \forall k,
$$

for some constant $a > 0$.

**Proposition J.2** (Expected diagonal under standard initialization). *For a LFCN with standard initialization, the expected path magnitude associated with a parameter in layer $k$ is given by*

$$
\mathbb{E}[g_i] = 
\begin{cases}
\dfrac{n_L}{n_{k+1}} a^{L-1-k}, & \text{if } i \text{ is a bias in layer } k, \\[1.5em]
\dfrac{n_L}{n_{k+1}} a^{L-1} + \dfrac{n_L}{n_{k+1}} \displaystyle\sum_{j=0}^{k-1} \dfrac{a^{L-1-j}}{n_j}, & \text{if } i \text{ is an edge in layer } k.
\end{cases}
\tag{94}
$$

### J.4. Consequences for the Rescaling Algorithm

For the rescaling algorithm to have a non-trivial effect, the expected diagonal of $G$ must exhibit variation across parameters. The analysis above reveals that this depends delicately on both the network architecture and the initialization scheme.

**Architectures with non-constant width.** We focus on edge parameters as their behavior already allows to highlight architectural regimes where the expected diagonal of $G$ exhibits significant variations in magnitude. For an edge parameter in layer $k$, equation (94) becomes

$$
\mathbb{E}[g_i] = \frac{n_L}{n_{k+1}} a^{L-1} + \frac{n_L}{n_{k+1}} \sum_{j=0}^{k-1} \frac{a^{L-1-j}}{n_j}.
$$

The behavior of this expression depends critically on the regime of the variance scaling factor $a$.

In the *large variance regime* ($a \to +\infty$), the first term dominates:

$$
\mathbb{E}[g_i] \sim \frac{n_L}{n_{k+1}} a^{L-1}.
$$

The expected path magnitude is determined primarily by the ratio $n_L / n_{k+1}$. For 2 parameters $i$ and $i'$ respectively in layers $k$ and $k'$, the ratio between $\mathbb{E}[g_i]$ and $\mathbb{E}[g_{i'}]$ is of order $n_{k'+1}/n_{k+1}$, provided that $a$ is large enough.

In the *small variance regime* ($a \to 0$), the sum dominates and retains only its last term:

$$
\mathbb{E}[g_i] \sim \frac{n_L}{n_{k+1}} \frac{a^{L-k}}{n_{k-1}}.
$$

This regime exhibits much stronger variation in expected path magnitudes. The ratio between $\mathbb{E}[g_i]$ and $\mathbb{E}[g_{i'}]$ for parameters in different layers can be of order $1/a$, which is very large.

In the *critical case* $a = 1$, corresponding for instance to Kaiming normal initialization with unit gain and fan-in scaling, we obtain

$$
\mathbb{E}[g_i] = \frac{n_L}{n_{k+1}} \left( 1 + \sum_{j=0}^{k-1} \frac{1}{n_j} \right), \quad k \in \{1, \dots, L-1\}.
$$

Hence, unless the width is constant across all layers, the expected diagonal of $G$ exhibits non-trivial variation.

## J.5. Constant-Width Networks

**Setup.** We now consider multi-layer perceptrons with constant width $n_k = n$ for all hidden layers $k \in \{0, \dots, L-1\}$. In this simplified setting, the expected path magnitude for an edge parameter in layer $k$ reduces to:

$$\mathbb{E}[g_i] = \begin{cases} a^{L-1}, & \text{if } k = 0, \\ a^{L-1} + \dfrac{1}{n}\displaystyle\sum_{j=1}^{k} a^{L-j}, & \text{if } k \in \{1, \dots, L-1\}. \end{cases} \tag{95}$$

**Case $a = 1$.** When $a = 1$, equation (95) simplifies to

$$\mathbb{E}[g_i] = 1 + \frac{k}{n}, \quad k \in \{0, \dots, L-2\}. \tag{96}$$

The expected path magnitude increases linearly with depth. For networks with large width $n \gg L$, the diagonal is approximately constant, and the rescaling algorithm has little effect.

**Case $a \neq 1$.** When $a \neq 1$, the geometric series yields

$$\mathbb{E}[g_i] = a^{L-1} + \frac{a^{L-1}}{n} \cdot \frac{1 - (1/a)^k}{1 - 1/a}, \quad k \in \{0, \dots, L-1\}. \tag{97}$$

**Large variance regime: $a \to +\infty$.** In this regime,

$$\mathbb{E}[g_i] \sim \begin{cases} a^{L-1}, & \text{if } k = 0, \\ a^{L-1}\big(1 + 1/n\big), & \text{if } k \in \{1, \dots, L-1\}. \end{cases} \tag{98}$$

**Small variance regime: $a \to 0$.** In this regime,

$$\mathbb{E}[g_i] \sim \begin{cases} a^{L-1}, & \text{if } k = 0, \\ a^{L-1}\big(1 + 1/n\big), & \text{if } k = 1, \\ \dfrac{a^{L-k}}{n}, & \text{if } k \in \{2, \dots, L-1\}. \end{cases} \tag{99}$$

The diagonal exhibits a multi-level structure with significant variation across layers, especially for deeper layers.

## J.6. Summary and Implications

Based on the analysis above, we identify regimes in which the rescaling algorithm is expected to have a non-trivial effect.

- **Varying-width architectures:** For networks with non-uniform layer widths, regardless of the initialization scale $a$, the expected diagonal of $G$ is non-constant across layers. The rescaling algorithm is expected to have a significant effect on such irregular configurations.

- **Near constant-width architectures with small variance:** For networks with approximately uniform width $n_i \approx n$ for all $i$, the algorithm has a non-trivial effect when initialized with small variance, i.e., when $a \ll 1$. In this regime, the diagonal exhibits significant variation across layers despite the regularity of the architecture.

- **Near constant-width architectures with critical initialization:** For such regular configurations with $a = 1$, the diagonal varies as $1 + k/n$. The effect is significant when the depth $L$ is comparable to or larger than the width $n$.

- **Near constant-width architectures with large variance:** For near constant-width networks with $a \gg 1$, the diagonal exhibits a simple two-level structure. The rescaling may help but the effect is limited to distinguishing the first layer from subsequent layers.

**Conclusion.** The rescaling algorithm is expected to have the most significant impact in two scenarios:

1. Varying-width architectures, regardless of initialization.

2. Near constant-width architectures of width $n$, initialized with small standard deviation relative to $1/\sqrt{n}$.

In both cases, the expected path magnitudes exhibit significant variation across parameters, leading to ill-conditioning of the path-magnitude matrix $G$ that the rescaling algorithm can address.

We empirically validate these assumptions through a controlled experiment on synthetic networks.

**Experimental Setup.** We generate multiple networks with a fixed depth of 8 layers and a mean width of 32 neurons per layer, ensuring a total budget of $8 \times 32 = 256$ neurons distributed across all hidden layers. To systematically vary the width regularity of the network architecture, we introduce a concentration parameter $\alpha \in [10^{-1}, 10^2]$ that controls a symmetric Dirichlet distribution used to sample the proportion of neurons allocated to each layer. When $\alpha$ is very large, the Dirichlet distribution becomes highly concentrated around uniform proportions, resulting in near constant-width configurations (all layers $\approx 32$ neurons). Conversely, when $\alpha$ is small, the distribution allows for greater variability in the proportions, creating varying-width architectures where some layers are significantly wider or narrower than others. To ensure architectural validity, each layer is guaranteed a minimum width of 1 neuron, with the remaining budget distributed proportionally according to the Dirichlet samples using a largest remainder method to maintain the exact total neuron count.

We repeat this experiment 20 times, sampling 20 different $\alpha$ values in each run to ensure statistical robustness.

**Remark.** We can notice that these favorable regimes are common in practice. Even when most hidden layers share a near constant-width, input and output dimensions often differ significantly, giving rise to an inherently non-uniform configuration. For instance, in the CIFAR-10 experiments (Figure 2), the 3072-dimensional input and 10-dimensional output create architectural irregularity despite the many intermediate layers of width 500. However, as depth increases, the network approaches near constant-width behavior, which explains why the performance gains decrease for very deep networks.

## K. Additional Experiments for Section 5.1

We provide here additional experiments complementing Section 5.1, covering a broader range of learning rates.

Figure 9 demonstrates that `PathCond` achieves strong performance at small to moderate learning rates, consistent with our theoretical framework where proper initialization critically affects training dynamics. However, performance degrades at larger learning rates, where optimization instabilities overshadow initialization benefits and our theoretical assumptions (small learning rate regime) no longer hold.

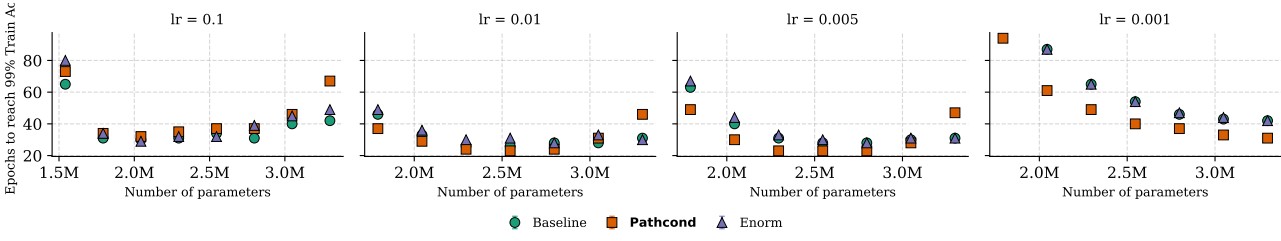

*Figure 9.* Convergence speed across different learning rates and network sizes.

## L. Additional Experiments for Section 5.2

We provide complementary experiments for Section 5.2, exploring the impact of learning rate on the generalization benefits of `PathCond` across a wider range of values.

Figure 10 presents training and test dynamics for learning rates ranging from $10^{-2}$ to $10^{-4}$. Consistent with our theoretical predictions, `PathCond` demonstrates improved convergence and generalization performance at small to moderate learning

rates, where proper initialization plays a critical role in training stability. For the smallest learning rate ($lr = 10^{-4}$), we extend training to 500 epochs to observe convergence behavior, while other configurations use 128 epochs.

The results confirm that `PathCond`'s advantages are most pronounced in regimes where our theoretical assumptions hold, with performance converging to baseline as learning rates increase and optimization dynamics begin to dominate over initialization effects.

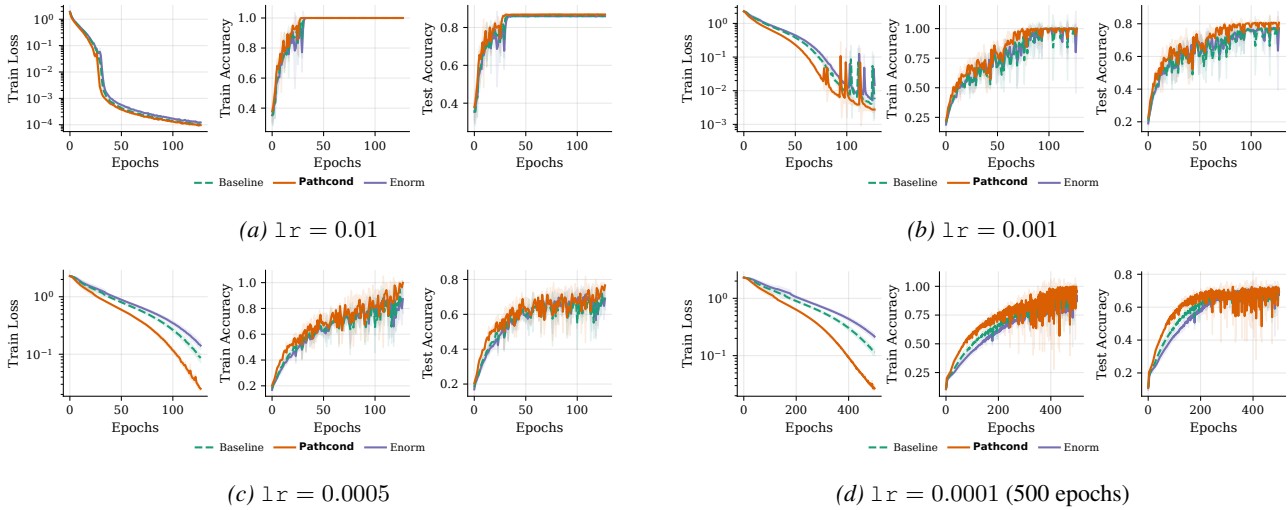

*(a)* `lr` $= 0.01$          *(b)* `lr` $= 0.001$

*(c)* `lr` $= 0.0005$          *(d)* `lr` $= 0.0001$ (500 epochs)

*Figure 10.* Training dynamics across different learning rates.

## M. Additional Experiments for Section 5.4

We provide complementary experiments for Section 5.4, exploring the impact of learning rate on the relationship between varying-width architectures and required rescaling adjustments.

Figure 11 presents the final training loss and rescaling magnitudes across different compression factors for three learning rates. Consistent with our theoretical framework, the required rescaling magnitude increases with architectural width variation (higher compression factors) across all learning rates.

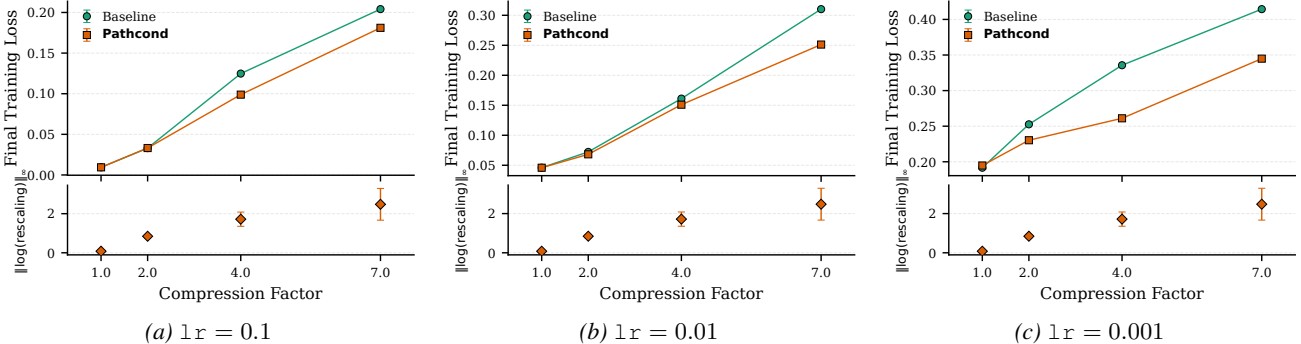

*(a)* `lr` $= 0.1$       *(b)* `lr` $= 0.01$       *(c)* `lr` $= 0.001$

*Figure 11.* Final training loss (top) and rescaling magnitude (bottom) across compression factors for different learning rates.

## N. `ENorm` : Experimental Setup

We describe the `ENorm` hyperparameter choices used in our comparisons. `ENorm` (Stock et al., 2019) rescales network weights at regular intervals to minimize a weighted sum of per-layer norms,

$$\sum_{\ell} c_{\ell} \|W_{\ell}\|_p^p$$

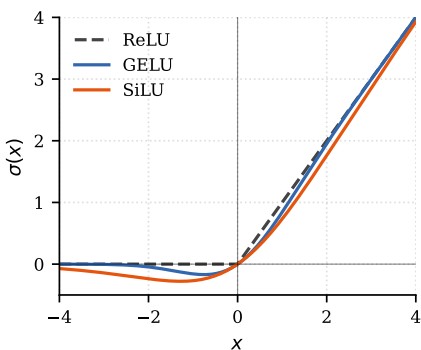
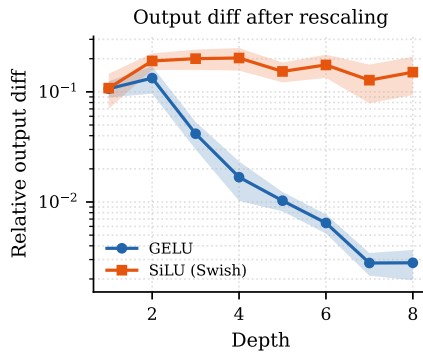

*Figure 12.* (**Left**) Smooth activation functions compared to ReLU. (**Right**) Approximation error of `PathCond` rescaling for MLPs with GELU and SiLU activations. Shaded bands show $\pm 1$ standard deviation over 10 random seeds.

where the depth-dependent coefficient $c_\ell$ affects each layers. We use the default value $p = 2$, so `ENorm` minimizes the sum of squared $\ell_2$ norms across layers. We set $c = 1$, which removes the depth penalty and treats all layers uniformly; as noted by the original authors, values of $c$ at or slightly above 1 generally yield the best results.

`ENorm` can be applied every $k \geq 1$ SGD iterations. Following the recommendation of the original authors, we apply one `ENorm` cycle after each SGD iteration ($k = 1$).

For networks with BatchNorm layers, we follow the recommendation of the authors and exclude BatchNorm parameters from the rescaling cycle, applying `ENorm` to the remaining weights only—consistently with how `PathCond` handles BatchNorm.

**In summary**, all `ENorm` results in this paper use the default configuration: $p = 2$, $c = 1$, one rescaling cycle per SGD iteration, BatchNorm layers excluded.

## O. Smooth Activation Functions

Strictly speaking, PathCond is designed for positively 1-homogeneous activation functions, such as ReLU or absolute value, for which the rescaling *exactly* preserves the input-output mapping of the network. For smooth, approximately homogeneous activations such as GELU (Hendrycks & Gimpel, 2016) and SiLU (or Swish) (Ramachandran et al., 2017), this invariance no longer holds exactly. Nevertheless, since GELU and SiLU both behave asymptotically like the identity (or zero) on the positive (respectively negative) part of the real line, the rescaling remains *approximately* input-output preserving at initialization. In this section, we empirically quantify this approximation error and show that, despite the loss of exact invariance, applying PathCond at initialization still accelerates training for such networks.

**Activation functions.** We recall the definitions of the two smooth activations considered here:

$$\text{GELU}(x) = x \, \Phi(x), \qquad \text{with } \Phi \text{ the standard Gaussian CDF,} \tag{100}$$

$$\text{SiLU}(x) = x \, \sigma(x), \qquad \text{with } \sigma(x) = 1/(1 + e^{-x}) \text{ the sigmoid.} \tag{101}$$

Both functions are smooth and approximate the ReLU on the positive part of the real line, with SiLU exhibiting a slightly larger negative dip than GELU. Functions are displayed on Figure 12 (**Left**).

**Experimental protocol.** We use the MLP architecture described in Section 5.1 (depths $L \in \{2, \ldots, 9\}$ with hidden width 500), replacing ReLU with either GELU or SiLU. For each depth, we draw the network parameters $\theta$ at initialization, apply the PathCond rescaling to obtain $\tilde{\theta}$, and measure the relative output difference

$$\mathcal{E}(x) = \frac{\|f_\theta(x) - f_{\tilde{\theta}}(x)\|}{\|f_\theta(x)\|}, \tag{102}$$

where $x$ is drawn from a standard Gaussian distribution $\mathcal{N}(0, I)$. We report $\mathcal{E}(x)$ averaged over 10 random initializations and input samples. We set $n_{\text{iter}} = 10$ and constrain the primal iterate to $\|u\|_\infty \leq 5$, which bounds the rescaling factors as $\lambda_h \in [e^{-5}, e^5]$. Without this constraint, the rescaling can produce NaNs in MLPs with GELU activations.

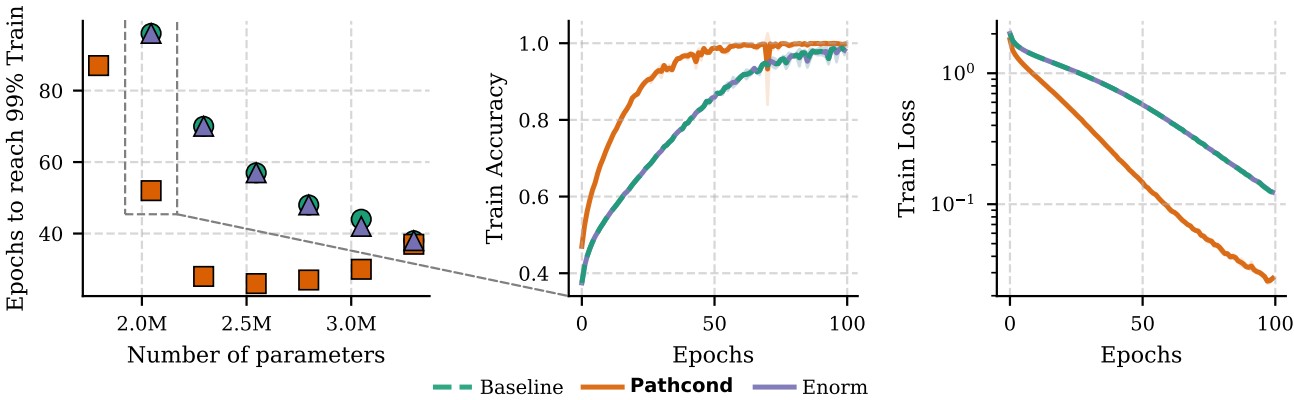

*Figure 13.* PathCond performance comparison across network depths on CIFAR-10 with multilayer perceptrons with GELU activation function. (**Left**) Number of epochs required to reach 99% training accuracy for networks with 2 to 8 hidden layers (abscissa = number of parameters, which increases with depth). (**Middle**) Training accuracy curves for the 3-hidden-layer network. (**Right**) Corresponding training loss curves

**Results.** Figure 12 (**Right**) reports the relative output difference $\mathcal{E}(x)$ as a function of depth for both activations. For GELU, the relative error *decreases* with depth, from approximately $10^{-1}$ at $L = 2$ down to $10^{-3}$ at $L = 9$. This behavior is consistent with the fact that GELU is closer to the identity than SiLU. Despite this non-negligible approximation error, applying PathCond at initialization still yields the same qualitative training speed-up observed for ReLU networks (see Figure 13). This suggests that exact input-output preservation is not strictly required for the rescaling to be beneficial: what matters is that the rescaled parameters land in a well-conditioned region of parameter space, which the PathCond procedure seems to achieve regardless of the activation's exact homogeneity.

