# OpenReview forum: "Path-conditioned training: a principled way to rescale ReLU neural networks"
_ICML.cc/2026/Conference — ICML 2026 regular_

### Official Review · Reviewer_h5Nw · 2026-03-02

**Soundness:** 3
**Presentation:** 3
**Significance:** 2
**Originality:** 3
**Overall Recommendation:** 4
**Confidence:** 2

**Summary:**

This paper proposes PathCond, a path-conditioned training procedure for ReLU networks that exploits rescaling invariance by teleporting parameters within an equivalence class without changing the realized function. It analyzes gradient dynamics in the lifted representation $z=\Phi(\theta)$, where training follows $\dot z=-P_\theta\nabla\ell(z)$, and designs a rescaling that makes $P_\theta\rightarrow I$ by minimizing a logdet-based divergence, yielding a convex objective. The resulting algorithm requires only $\mathrm{diag}(G_\theta)$ rather than full path objects. Empirically, the evaluation focuses on a flattened MLP and a small fully convolutional network on CIFAR-10, reporting faster convergence in terms of epochs and improved efficiency trends under the proposed rescaling.

**Compliance With Llm Reviewing Policy:**

Affirmed.

**Key Questions For Authors:**

1. How does PathCond interact with normalization layers (BatchNorm/LayerNorm) during training? Will normalization layers disrupt the simple positive homogeneity? Can you characterize when the rescaling invariance still holds or how to adapt the method?
2. Can PathCond expand to attention-based layer? If can , how?

**Limitations:**

yes

**Strengths And Weaknesses:**

Strengths:
- Introduces a new, geometry-motivated rescaling criterion derived from aligning the path-kernel with the identity via spectral Bregman divergences
- Clever reduction from an intractable path-space object ($P_\theta ∈ \mathbb{R}^{q\times q}$) to a tractable parameter-space object ($G_\theta\in\mathbb{R}^{p\times p}$) using spectral divergence invariance and the admissible rescaling group.
- Demonstrates acceleration of training on CIFAR-10 for fully-connected models and a small conv net, with gains of up to 1.5× fewer epochs to target accuracy and no observed degradation relative to baseline and Equinormalization.

Weaknesses:
1. The paper motivates PathCond by aiming to make the lifted dynamics $\dot z=-P_\theta\nabla \ell(z)$ closer to the ideal $\dot z=-\nabla \ell(z)$, but it does not provide a clear, formal rationale for why making “ $P_\theta\rightarrow I$” is  a provable link from the proposed divergence minimization to guaranteed training acceleration. The current justification is quite intuitive.
2. The experiments omit natural rescaling-aware comparators explicitly discussed in related work, such as Path-SGD and G-SGD, leaving it unclear how PathCond performs relative to established symmetry-aware optimizers.
3. Empirical evaluation is restricted to a flattened MLP and a small fully convolutional network on CIFAR-10. The absence of training results on modern skip-connected architectures (e.g., ResNet) weakens the evidence for broad applicability.
4. Key results are reported over only 3 independent runs. Increasing to at least 10 independent runs would strengthen robustness claims.

---

> ### Author Rebuttal · Authors · 2026-03-31
>
> ## W1. Rationale for PathCond ?
> > *"The current justification [of PathCond] is quite intuitive."*
>
> A standard "natural gradient" viewpoint (Amari 1998, Martens 2020) is that GD directly in function space is a good criterion, but it is also known to be computationally hardly tractable. Lifted space is an intermediate space that preserves some interesting properties of function space, so our take is that GD in this space (with respect to a well chosen metric) is expected to be a good criterion. We believe our experiments confirm that it is indeed a better criterion than GD in parameter space.
>
> ## W2. Comparison to symmetry-aware optimizers ?
> > *"The experiments omit natural rescaling-aware comparators explicitly discussed in related work, such as Path-SGD and G-SGD, leaving it unclear how PathCond performs relative to established symmetry-aware optimizers."*
>
> On the toy example of Figure 1, we observed that Path-SGD (run with the code provided by the authors ([code PathSGD](https://github.com/bneyshabur/path-sgd/tree/master)) behaves essentially as the baseline, while PathCond improves over the baseline.
> The experiments corresponding to Figure 2 confirm this trend and will be added in the final version.
> Note also that while being symmetry-aware is desirable, it is not sufficient to ensure good behavior of an optimizer: choosing a rescaling-equivalent parameter according to ***any criterion*** leads to a symmetry-aware optimizer. The distinct behaviors of Path-SGD and PathCond on the toy example of Figure 1 highlight this phenomenon.
>
> ## W3. What about modern skip-connected architectures ?
> > *"Empirical evaluation is restricted to a flattened MLP and a small fully convolutional network on CIFAR-10. The absence of training results on modern skip-connected architectures (e.g., ResNet) weakens the evidence for broad applicability."*
>
> We did run the proposed method on ResNets-18 with learnable skip connections and small learning rate, **reaching the same computational benefits in terms of reduction of the number of epochs needed to reach a target train loss/accuracy** with SGD. We would be happy to include these in the final version if the reviewers find it useful. It is however known that the SGD optimizer with small learning rate is unfavorable to reach good test accuracy on such an architecture, and future work on combining PathCond with more advanced optimizers such as Adam, Shampoo, or Muon, would be desirable to explore these aspects.
>
> ## W4. Number of runs
> > *"Key results are reported over only 3 independent runs. Increasing to at least 10 independent runs would strengthen robustness claims."
>
> We acknowledge this limitation. We will run all experiments with 10 independent seeds in the final version. We note that the variance across our 3 runs is already low (reported in the paper), suggesting the results are stable.
>
> ## W5. Normalisation Layers
>
> > *"How does PathCond interact with normalization layers (BatchNorm/LayerNorm) during training? Will normalization layers disrupt the simple positive homogeneity? Can you characterize when the rescaling invariance still holds or how to adapt the method?"*
>
> **At initialization**, BN layers are initialized as the identity ($\gamma = 1$, $\beta = 0$, running statistics not yet accumulated), so they do not break the rescaling symmetry, PathCond applies without modification.
>
>
> **During training**, BN is a linear layer with bias and identity activation, so the rescaling can be propagated through it by rescaling the running mean and the bias accordingly, as shown in [Gonon et al. 2024](https://arxiv.org/pdf/2405.15006). Since all our experiments apply PathCond **once at initialization**, this is not needed in the current paper, but provides a clear path to extend PathCond to mid-training rescaling.
>
> ## W6. Extension beyond ReLU ? Transformers ?
>
> > *"Can PathCond expand to attention-based layer? If can , how?"*
>
> As in the answer to 8K2V "Q2. Beyond ReLU?": Attention layers also have some internal rescaling invariances, although they are more tricky due to the softmax. It seems feasible indeed to extend the PathCond philosophy to this setting, this is an interesting question for future work.

---

> > ### Author Rebuttal · Reviewer_h5Nw · 2026-04-03
> >
> > The rebuttal provides useful additional clarification, especially on BN-at-initialization and some preliminary evidence on Path-SGD and ResNet-18. However, my main conceptual concern remains: the link between making the path-kernel close to the identity and provable training acceleration is still largely heuristic. In addition, the comparison with symmetry-aware baselines remains incomplete (notably G-SGD), the promised 10-seed evidence is not yet shown, and the discussion on LayerNorm/attention remains mostly future-work level. Therefore I keep my score unchanged at 4.

---

### Official Review · Reviewer_jsTi · 2026-03-09

**Soundness:** 4
**Presentation:** 2
**Significance:** 2
**Originality:** 3
**Overall Recommendation:** 4
**Confidence:** 3

**Summary:**

The paper introduces a method to use the rescaling symmetries of ReLU networks to improve training and initialization stability. As ReLUs are positively homogeneous, the authors use a "path-lifting" framework to view this symmetry geometrically. They propose an algorithm to find an optimal rescaling of parameters at initialization that aligns the network's kernel, effectively preconditioning the gradient descent dynamics to speed up training. The authors present numerical results on FC and convolutional networks with ReLU activations and explore the regimes where the method performs best.

**Compliance With Llm Reviewing Policy:**

Affirmed.

**Final Justification:**

I am updating my score to Weak Accept because the author rebuttal successfully clarified my core concerns. I have kept my recommendation as "Weak" because, while the GELU, the theoretical framework remains relatively limited to ReLU. Additionally, while the authors argue that PathCond is complementary to modern second-order optimizers like Shampoo or Muon, a lack of experiment makes it difficult to fully quantify the relative significance of the gains in a modern training stack.

**Key Questions For Authors:**

Below are my questions:
1. Does the proposed setting and method extend to contemporary activation functions?
2. Did the authors experiment with contemporary architectures, such as Transformer networks?
3. More of a comment - the sentence in the abstract "We introduce a geometrically motivated criterion to rescale neural network parameters which minimization leads to..." reads odd.
4. The paper would benefit from a dedicated comparison of computational complexity w.r.t existing optimizers, like Adam or RMSProp. If the proposed "efficient alignment algorithm" requires computing Jacobians or large matrix decompositions, the wall-clock time might negate any reduction in training steps.
5. Regarding the preconditioner $P_{\theta}$, the method appears to prefer a full-rank Jacobian. If the Jacobian loses rank, it seems the preconditioner would collapse dimensions, potentially trapping the optimizer and preventing it from exploring the full function space. Given this, could the authors clarify how their method interacts with the concept of sparsity? Specifically, does a full-rank Jacobian ensure smooth transition in the lifted space inherently conflict with finding sparse representations?

**Limitations:**

The proposed method's strict reliance on the positive homogeneity of ReLU activations limits its immediate applicability to modern architectures, such as Transformers, which utilize non-homogeneous activations (e.g., GELU) and normalization layers. Additionally, the computational overhead required to calculate the alignment criterion and Jacobians may not be competitive against standard optimizers like Adam. Furthermore, relying on "small variance" initializations risks triggering the well-documented "dying ReLU" problem.

**Strengths And Weaknesses:**

**Strengths:**
1. Theoretical rigor - the paper proposes a sound mathematical characterization of the proposed framework, connecting rescaling symmetry with path conditioning and gradient flows. The claims are backed with theoretical statements.
2. Novelty - The authors propose a novel outlook on network initialization - rather than considering initialization as a ‘weight sampling’ strategy, the authors think of proper initialization as an alignment problem of the initial tangent kernel.

**Weaknesses:**
1. Architectural relevance - The paper focuses on ReLU networks, FC ( and convolutional) networks. However, in most cases, contemporary networks comprise more contemporary components, both on the architecture level (e.g. Transformer networks) and non-ReLU activations (e.g., GeLU, SwiGLU).
2. Missing baselines - There is existing work on preconditioners (such as Shampoo, Muon). However, those are not mentioned in the paper and not compared to. This makes it harder to contextualize the paper within existing literature and to clearly quantify its benefits over existing works.
3. Small variance issue - The authors claim the method performs well in small variance environments. However, initializing ReLU networks with very small variance notoriously leads to "dead ReLUs" [1] (where activations permanently output zero) and vanishing gradients.

[1] Lu, Lu, et al. "Dying relu and initialization: Theory and numerical examples." arXiv preprint arXiv:1903.06733 (2019).

---

> ### Author Rebuttal · Authors · 2026-03-31
>
> ## W1 — Extension beyond ReLU ? Transformers ?
> > *"Does the [...] method extend to contemporary activation functions?"*
> > *"Did the authors experiment with contemporary architectures, such as Transformer networks?"*
>
> Please refer to the detailed answer made to 8K2V: "Q2. Beyond ReLU?".
>
> ## W2 — Relation to Shampoo, Muon ?
>
> > *"There is existing work on preconditioners (such as Shampoo, Muon). However,
> > those are not mentioned in the paper and not compared to."*
>
>
> Shampoo and Muon can be seen as second-order optimizers that, like Adam, modify gradient updates at each step by leveraging the matrix structure of per-layer weight parameters. We thank the reviewer for this suggestion and will include them in our discussion alongside Adam in the conclusion.
>
> We note, however, that these optimizers operate at a quite different level compared to PathCond, making direct comparison with PathCond difficult. We believe they are in fact complementary: while second-order optimizers act on the gradient update rule, PathCond takes a holistic geometric view of the network parameters through path-lifting, rescaling them — including biases, skip connections, and potentially on general DAGs — primarily at initialization.
> Combining Shampoo/Muon/Adam with the rescaled initialization of PathCond would be an interesting step for future work. We will clarify these points in the final version.
>
> ## W3 — Small variance / dying ReLU
>
> > *"Initializing ReLU networks with very small variance notoriously leads to
> > dead ReLUs."*
>
> As far as we know, Lu et al. (2019) show that the *born-dead* phenomenon occurs for **small network width** (Figure 2 of the paper) under standard He initialization — not for small initialization variance.
> Moreover, the *small-variance regime* we identify as favorable **does not require pathologically small variances: the default Kaiming uniform itself falls in this favorable regime.**
> Indeed, all our experiments use the default PyTorch Kaiming uniform initialization. At width $= 500$, dying ReLUs are both theoretically and empirically absent, and the "small" variance regime is reached.
>
> ## W4 — Computational overhead and timing costs
> > *"If the efficient alignment algorithm requires computing Jacobians or large
> > matrix decompositions, the wall-clock time might negate any reduction in
> > training steps."*
>
> Let us emphasize that, as detailed in Section 4.3, PathCond **does not require computing any full Jacobian or matrix decomposition**. We have indeed shown that, using the logdet divergence, we can build a criterion that only depends on the diagonal elements of the path kernel, which costs  $O($ \# $\mathrm{params})$ to compute, and can be calculated with a single forward pass.
> Our algorithm to solve the resulting optimization problem is also quite efficient. With our current implementation, we measured wall-clock time on an RTX 3090:
>
> | Architecture     | Params      | Rescaling | 1 Epoch CIFAR-10 | Ratio  |
> |------------------|:-----------:|:---------:|:----------------:|:------:|
> | MLP-GELU (3×500) | 2,042,510   | 2.14s     | 5.47s            | 0.39×  |
> | FullyConv        | 2,618,378   | 2.67s     | 11.04s           | 0.24×  |
> | ResNet-18C       | 11,528,266  | 5.11s     | 12.21s           | 0.42×  |
>
> Thus the rescaling costs less than half a training epoch and is performed **once**
> at initialization — the overhead over a full training run is thus negligible.
> Further significant optimizations appear feasible by parallelizing block coordinate descent along even / odd layers, and will be the object of future investigations.
>
> ## W5 — Sparsity / full-rank Jacobian
> > *"The method appears to prefer a full-rank Jacobian. Could the authors clarify
> > how their method interacts with sparsity?"*
>
> We assume that you are referring to the choice of a criterion that favors proximity between $P_\theta$ and the identity matrix. However, as detailed in the appendix, optimizing the rescaling with such a criterion simply promotes the choice of rescaling where $P_\theta$ is closest to a projection matrix of the same rank, but has no effect on this rank. The criterion is thus somehow oblivious to the rank. It  does not promote sparsity, but does not conflict with sparsity either.
>
> ## Minor — Abstract phrasing
> > *"The sentence '...criterion to rescale neural network parameters which
> > minimization leads to...' reads odd."*
>
> We thank the reviewer for catching this and will revise the phrasing.

---

> > ### Author Rebuttal · Reviewer_jsTi · 2026-04-03
> >
> > I thank the reviewers for their response to my points and concerns, and I consequently decided to raise my score.

---

### Official Review · Reviewer_nrxt · 2026-03-10

**Soundness:** 3
**Presentation:** 3
**Significance:** 3
**Originality:** 4
**Overall Recommendation:** 5
**Confidence:** 3

**Summary:**

The authors identify the problem of choosing optimal per-parameter rescalings for a ReLU neural network under the constraint of pathwise invariances, that is, rescaling network parameters such that the overall function implemented by the network is unchanged, and yet the  gradient flow training dynamics differ significantly. They leverage the path-lifting framework and propose a principled optimal scaling based on minimizing a Bregman matrix divergence between the so-called *path kernel*, which governs training dynamics in lifted space, and the identity, thereby approximating gradient descent in lifted space. They derive an efficient algorithm from this principle, and demonstrate gains in training speed without compromising generalization. Finally, they hypothesise and validate favourable regimes for the algorithm.

**Compliance With Llm Reviewing Policy:**

Affirmed.

**Final Justification:**

I was originally quite positive about the contributions of this paper, and the rebuttal addressed all of my major concerns. I still think it could be better motivated as to why GD in lifted space is desirable, but intuitively this feels right and the empirical results also support this. As such, I advocate for acceptance.

**Key Questions For Authors:**

- While the theory is derived exclusively for ReLU networks (for which the path-lifting framework is well-defined), I see no reason why one couldn't apply rescaling at initialization to a network using a ReLU-like nonlinearity (such as GELU) exactly as would be done for an equivalent network using ReLU. Would the authors be willing to add a small ablation repeating one of the experiments for non-ReLU networks to empirically verify usefulness in more general settings? Similarly, if the authors added one or two more larger-scale experiments then that would strengthen the paper considerably.
- Would the authors be willing to add a figure demonstrating the rescaling itself in parameter space, as mentioned in the presentation weaknesses section?
- How does the proposal interact with other modern parametrization research? For example, $\mu P$, NTP etc. Does the rescaling break the induced kernel or feature-learning regimes? Does the rescaling change the inductive biases of the networks somehow? And how is the choice of optimizer likely to affect the optimal rescaling?
- While it is clear empirically that this gives good results, why is it sufficient for this method to rescale only at initialization, while the quoted literature suggests that "teleportation" should be performed at every optimization step? Is it possible to derive bounds on the divergence of the path kernel from its initialization?
- Why is it expected that training speed should improve when training becomes close to gradient descent in lifted space? The conditioning argument is convincing to me, but I don't immediately see why this is desirable.

**Limitations:**

No - missing impact statement

**Strengths And Weaknesses:**

# Strengths #
- Soundness: The underlying idea is principled, and is supported by a theoretical backbone from the literature and substantial analysis in the appendix. Some experiments are provided demonstrating the algorithm and its improvements in training speed, and importantly, at least as good test metrics. The analysis on favourable regimes is a nice bonus and strengthens the paper.
- Presentation: The paper reads very nicely, with a good mixture of equations and figures etc. The logical flow is generally good, and the idea is presented in an easy to understand manner. Most of the relevant literature is discussed and the proposed method is sufficiently placed in the literature. The framing as a conditioning improvement problem by the choice of Bregman divergence is very compelling.
- Significance: Understanding training dynamics and optimal parametrizations in general is of high importance and significant research interest over the last few years, and this paper proposes a meaningful improvement in these areas. Moreover, it is evident the authors have paid significant attention to the efficiency of the proposed algorithm which is likely to help with adoption.
- Originality: The proposed rescaling criterion is, to the best of my knowledge, novel. Moreover, the objective of achieving gradient descent in lifted space is interesting, and the work of this paper may form a good basis for future analysis of training dynamics in this space which could be quite fruitful, especially when paired with efficient implementations as proposed here.

# Weaknesses #
- Soundness: Ideally I would've liked to see a wider base of experiments across a range of architectures, for example on a small modern transformer. I would've also liked to see some analysis on the effect of underlying parametrization scheme (for example $\mu P$ Yang and Hu 2021) on the performance of the rescaling algorithm, and experiments applying the rescaling algorithm to non-ReLU networks. See the below questions for a more detailed explanation.
- Presentation: A figure demonstrating the rescaling in a low-dimensional parameter space model would be an informative addition and really strengthen the paper. Additionally, there are a few sentences which don't make sense: "...criterion to rescale neural network parameters **which** minimization leads to..." (abstract) would maybe read better replacing "which" with "whose", "First, it has proven useful **in understanding** existing..." (Page 1 right column second paragraph) would maybe read better replacing "in understanding" with "to understand", and ""teleport" any given parameter $\theta$ **by** a rescaling-equivalent one $\theta'$" would read better replacing "by" with "to". Finally, the required "Impact Statement" is missing
- Significance: As mentioned above, the direct applicability of the theory to ReLU networks only limits its utility in modern networks. This, of course, could be easily remedied by empirical evaluation of the algorithm on non-ReLU networks.
- Originality: No meaningful originality weaknesses.

Yang, Greg, and Edward J. Hu. "Tensor programs iv: Feature learning in infinite-width neural networks." International Conference on Machine Learning. PMLR, 2021.

---

> ### Author Rebuttal · Authors · 2026-03-31
>
> ## Q1. Beyond ReLU ? Larger-scale ?
> > *"[...] Would the authors be willing to add a small ablation repeating one of the experiments for non-ReLU networks [...] ? Similarly, if the authors added one or two more larger-scale experiments then that would strengthen the paper considerably."*
>
> **Non-ReLU activations.** As detailed in our response to Reviewer 8K2V: "Q2. Beyond ReLU?", we applied PathCond to MLP-GELU networks on CIFAR-10. The output is quasi-preserved after rescaling and PathCond achieves the same convergence gains as on ReLU for a 3 hidden layers architecture: $\sim 50$ epochs to reach 99% train accuracy vs. $\sim 100$ for the baseline.
>
> **Larger-scale experiments.** As detailed in our response to Reviewer h5Nw (W3. What about modern skip-connected architectures ?), we trained a ResNet-18 with learnable skip connections using PathCond, reaching the same computational benefits in terms of reduction of the number of epochs needed to reach a target train loss/accuracy with SGD.
>
> ## Q2. Illustration ?
> > *"Would the authors be willing to add a figure demonstrating the rescaling itself in parameter space?"*
>
> Thank you for the suggestion. We would like to draw the reviewer's attention to Figure 1 of the paper, which exactly illustrates this point:  it shows the rescaling orbit in parameter space $\Theta$ alongside the corresponding trajectory in the lifted space $\Phi(\Theta)$, for a toy 1-hidden-layer ReLU network.
>
> ## Misc.
> > *"There are a few sentences which don't make sense. Finally, the required > Impact Statement is missing."*
>
> We thank the reviewer for catching these. We will fix the pointed sentences and add the missing Impact Statement in the final version.
>
> ## Q3. Interaction with $\mu P$ etc. ?
> > *"How does the proposal interact with other modern parametrization research? For example, $\mu P$ , NTP etc. Does the rescaling break the induced kernel or feature-learning regimes? Does the rescaling change the inductive biases of the networks somehow? And how is the choice of optimizer likely to affect the optimal rescaling?”*
>
>
> As the rescaling only changes the parameters but not the implemented function, it does not change any kernel based on the function itself. However, kernels defined in terms of *gradients (with respect to parameters)* of the function, such as the NTK, are indeed changed, and we expect that the corresponding inductive biases are also somehow changed. On the examples considered in this paper, the empirical impact on test accuracy is either negligible or positive, suggesting that the possibly modified inductive bias is somehow favorable. Should one encounter settings where this is no longer the case, an interesting perspective would be to inject a more explicit version of the "right" inductive bias via a regularization term, while still exploiting the PathCond philosophy to geometrically speedup training.
>
> Regarding $\mu P$, please see also our answer to 8K2V "Q1. Connection to $\mu P$ ?"
>
> Regarding the interplay with the optimizer, nothing prevents the use of the PathCond rescaling with a different optimizer. That said, an interesting perspective would to be to design alternative optimizer-specific rescaling criteria, leveraging the ODE associated to an optimizer in the same way gradient flow is used to derive PathCond.
>
>
> ## Q4. To rescale at every step or not ?
> > *"Why is it sufficient for this method to rescale only at initialization, while the quoted literature suggests that "teleportation" should be performed at every optimization step? Is it possible to derive bounds on the divergence of the path kernel from its initialization?”*
>
> First, we find it an interesting behavior that (unlike previous methods such as E-norm) rescaling only at initialization can have a significant effect. We also did run experiments with periodic rescaling using PathCond. Since we observed that most of the rescaling effect appeared at init, for clarity of exposition we chose to focus on rescaling at init in our presentation. If the reviewers think it is useful we could of course easily add the corresponding ablation study in the final version. An interesting perspective for future work would be to decipher the reasons why the PathCond rescaling seems to vary slowly along training, and how this may depend on architecture or initialization scale.
>
> ## Q5. Rationale for PathCond ?
> > *"Why is it expected that training speed should improve when training becomes close to gradient descent in lifted space?”*
>
> A standard "natural gradient" viewpoint (Amari 1998, Martens 2020) is that GD directly in function space is a good criterion, but it is also known to be computationally hardly tractable. Lifted space is an intermediate space that preserves some interesting properties of function space, so our take is that GD in this space (with respect to a well chosen metric) is expected to be a good criterion. We believe our experiments confirm that it is indeed a better criterion than GD in parameter space.

---

> > ### Author Rebuttal · Reviewer_nrxt · 2026-03-31
> >
> > I thank the authors for the additional experiments and interesting discussion. My concerns have been mostly addressed, and I will thus increase my score to 5.
> >
> > Thank you also for pointing me to the referenced part of Figure 1. As an aside, which I do not intend to impact the review and do not require to be done at all, I think it would be interesting to also include a plot of a path conditioned and standard network trained on some very simple toy problem to help understand the impact on network inductive bias. This could be something so simple as regressing to the point of interpolation onto a tiny dataset generated from a noised 1d function, e.g. $y_i=\sin(x_i)+w_i$ for some normally distributed $w_i$ with ~5 datapoints, which would also be very easy to plot. If the authors have time, I encourage them to consider something like this to improve the already excellent presentation of the paper even further.

---

### Official Review · Reviewer_8K2V · 2026-03-15

**Soundness:** 4
**Presentation:** 4
**Significance:** 3
**Originality:** 3
**Overall Recommendation:** 5
**Confidence:** 3

**Summary:**

It is well known that ReLU neural networks exhibit rescaling invariance. However, such rescaling can significantly alter the training dynamics of the model. This paper proposes a method to rescale neural network parameters according to a geometrically motivated criterion based on the path-lifting framework. The authors derive an efficient algorithm for this procedure and conduct small-scale experiments to demonstrate the resulting improvements.

**Compliance With Llm Reviewing Policy:**

Affirmed.

**Final Justification:**

My major concerns have been addressed, and my original rating has already reflected my positive rating.

**Key Questions For Authors:**

1. Connections to µP: Can the authors comment on the relationship between this framework and the maximal update parameterization (µP) framework (https://proceedings.mlr.press/v139/yang21c/yang21c.pdf), as well as related spectral norm scaling conditions (https://arxiv.org/abs/2310.17813), particularly in the setting of unbalanced layer widths?
2. Beyond ReLU: To what extent can this framework extend beyond ReLU networks? For instance, transformers with layer normalization exhibit rotational symmetries (https://arxiv.org/pdf/2401.15024). Could the proposed framework potentially address such symmetries?

**Limitations:**

Yes

**Strengths And Weaknesses:**

Strengths:
1. Soundness: The technical contributions of this paper are very solid. The entire framework is well formulated with clean notation and sufficient proofs provided in the appendix. All procedures are described with clear algorithmic steps, and the claims are well supported.
2. Presentation: The simple one-dimensional examples and illustrative figures are very helpful, and the presentation is well paced and clear. It is impressive that the authors have turned this technically dense paper into something enjoyable to read.

Weaknesses:
Significance: This work mainly focuses on the rescaling invariance of ReLU neural networks. While this is definitely interesting, the paper does not address other types of parameterization symmetries that arise in neural networks. Furthermore, ReLU is no longer the default choice for constructing neural networks, as many other activation functions are now widely used.

---

> ### Author Rebuttal · Authors · 2026-03-31
>
> ## Q1. Connection to $\mu P$ ?
> > *"Can the authors comment on the relationship between this framework and the maximal update parameterization (µP) framework (https://proceedings.mlr.press/v139/yang21c/yang21c.pdf), as well as related spectral norm scaling conditions (https://arxiv.org/abs/2310.17813), particularly in the setting of unbalanced layer widths?”*
>
> Thank you for pointing out the possible connections with these interesting references on the role of initialization scales/learning rates and their compatibility with kernel regimes and feature learning abilities. Practically speaking, PathCond is complementary, in the sense that it could be empirically applied to rescale the initialization provided by such frameworks. An interesting perspective would notably be to observe the shape of the obtained rescaling, or even to predict this shape using the approach from our analysis of initialization regimes. A discussion on these aspects will definitely be added in the final version.
> It would also be particularly interesting to understand whether the geometric principle behind PathCond, which is applicable to general DAG architectures, recovers the spectral condition heuristic of the second reference.
>
>
> ## Q2. Beyond ReLU?
> > *"To what extent can this framework extend beyond ReLU networks? For instance, transformers with layer normalization exhibit rotational symmetries (https://arxiv.org/pdf/2401.15024). Could the proposed framework potentially address such symmetries?”*
>
> Strictly speaking, PathCond is designed for  positively 1-homogeneous activation functions. For approximately homogeneous activations (GELU, SiLU), rescaling invariance is only approximate, yet it is possible to exploit PathCond at initialization. It (slightly) modifies the input-output mapping at initialization, and we conducted experiments showing that it indeed still speeds up training essentially as with ReLU. Quantitatively, rescaling GeLU/SiLU networks at initialization using PathCond preserves the input-output mapping up to <10% relative error, while on CIFAR-10 with a 3-hidden layer MLP-GELU, training takes 50 epochs to reach 99% train accuracy compared to ~100 epochs for the baseline without PathCond rescaling. These results will be included in the final version.
> Attention layers also have some internal rescaling invariances, although they are more tricky due to the softmax. It seems feasible indeed to extend the PathCond philosophy to this setting.
> Another possible approach would be to replace the softmax with a ReLU activation, as proposed in [A], which would allow PathCond to be applied more directly. In any case, the adaptation of PathCond to attention layers is an interesting avenue for future work.
>
> [A] Replacing softmax with relu in vision transformers, M Wortsman, J Lee, J Gilmer, S Kornblith, 2023.

---

> > ### Author Rebuttal · Reviewer_8K2V · 2026-04-03
> >
> > The authors show that, although rescaling invariance is only approximate, the proposed approach can still accelerate training for GeLU and SiLU. The connection with $\mu P$ is not explored in depth, but this lies beyond the scope of the paper.

---

### Decision · Program_Chairs · 2026-04-30

**Decision:**

Accept (regular)

**Comment:**

The authors propose PathCond, a rescaling strategy for ReLU networks that exploits their inherent rescaling invariance to accelerate training. Building on the path-lifting framework, they show that different parameter rescalings, while functionally equivalent, induce different geometries via a path-kernel that governs training dynamics in the lifted representation. The optimal rescaling corresponds to a linear local geometry, and an efficient algorithm is proposed to compute it. Empirical results demonstrate improved training performance under the proposed method.

The proposed approach is technically sound and constitutes a solid contribution, with a well-formulated framework and claims that are well supported. The paper is clearly written and accessible, making the underlying ideas easy to follow. The method itself is elegant and principled, and is supported by both theoretical analysis and empirical validation. The authors position their work appropriately within the existing literature, providing a relevant and well-contextualized discussion. The problem addressed is timely, and the proposed solution is both novel and insightful, particularly in its formulation of optimization in the lifted space. Additionally, the interpretation of initialization as an alignment problem of the tangent kernel offers a new interesting perspective.

The work primarily focuses on ReLU networks, and although some results are provided for related activations, the scope of the parameterization symmetries considered remains somewhat limited. The experimental validation could be more extensive to better support the generality of the claims. Furthermore, a more explicit discussion and comparison with modern preconditioning methods would strengthen the paper and help place the contribution even more clearly within the broader optimization literature. Finally, the choice of the proposed optimality criterion for rescaling, while intuitive, could benefit from a justification or deeper discussion.

The concerns raised by the reviewers have been addressed to a significant extent during the rebuttal phase, which is appreciated. Overall, the paper represents a meaningful contribution to a timely problem, proposing an elegant and principled solution supported by both theoretical and empirical results. For these reasons, I recommend acceptance, while encouraging the authors to incorporate the reviewers’ feedback to further strengthen the paper.